# E-cardiac patch to sense and repair infarcted myocardium

**Renjie Qiu**[1,2,3,10], **Xingying Zhang**[4,10], **Chen Song**[5,10], **Kaige Xu**[4], **Huijia Nong**[1], **Yi Li** ®[6], **Xianglong Xing**[1], **Kibret Mequanint** ®[7], **Qian Liu**[8], **Quan Yuan** ®[9], **Xiaomin Sun**[2], **Malcolm Xing** ®[4] ✉ & **Leyu Wang** ®[1,3] ✉

Conductive cardiac patches can rebuild the electroactive microenvironment for the infarcted myocardium but their repair effects benefit by carried seed cells or drugs. The key to success is the effective integration of electrical stimulation with the microenvironment created by conductive cardiac patches. Besides, due to the concerns in a high re-admission ratio of heart patients, a remote medicine device will underpin the successful repair. Herein, we report a miniature self-powered biomimetic trinity triboelectric nanogenerator with a unique double-spacer structure that unifies energy harvesting, therapeutics, and diagnosis in one cardiac patch. Trinity triboelectric nanogenerator conductive cardiac patches improve the electroactivity of the infarcted heart and can also wirelessly monitor electrocardiosignal to a mobile device for diagnosis. RNA sequencing analysis from rat hearts reveals that this trinity cardiac patches mainly regulates cardiac muscle contraction-, energy metabolism-, and vascular regulation-related mRNA expressions in vivo. The research is spawning a device that truly integrates an electrical stimulation of a functional heart patch and self-powered e-care remote diagnostic sensor.

The electroactive property of the myocardium enables synchronous electrical signal transmission and rhythmic heartbeat[1]. When myocardial infarction (MI) occurs, due to the weak regeneration ability of cardiomyocytes (CMs), the infarcted myocardium exhibits a blind region in electroactivity with the consequent remodeling of electrophysiology and structure, which is the main culprit behind a sudden cardiac death[2,3]. Due to the insufficiency in current treatments of pharmacological therapy and bypass operation for MI, and the scarcity of donors for heart transplantation, the morbidity and mortality of MI

patients demonstrate a tendency to escalate. In the last two decades, the spark ignited by cardiac patches (CPs), which mimic heart composition, has been significant enough to fascinate researchers as a promising strategy for cardiac repair after MI[4]. Our previous studies have proved that conductive CPs (CCPs) can reconstruct the electroactive microenvironment[5,6], which plays a crucial role in restoring the cardiac function of the infarcted heart. Our previous injectable and highly conductive CPs have made significant progress in infarcted hearts of minipigs[5]. However, there are significant hurdles to overcome

[1]Guangdong Provincial Key Laboratory of Construction and Detection in Tissue Engineering; Biomaterials Research Center, School of Biomedical Engineering, Southern Medical University, Guangzhou, Guangdong, China. [2]School of Traditional Chinese Medicine, Southern Medical University, Guangzhou, Guangdong, China. [3]Department of Anatomy, School of Basic Medical Sciences, Guangzhou Medical University, Guangzhou, Guangdong, China. [4]Department of Mechanical Engineering, University of Manitoba, Winnipeg, MB, Canada. [5]The Fifth Affiliated Hospital of Southern Medical University, Southern Medical University, Guangdong, Guangzhou, China. [6]Department of Biochemistry and Molecular Biology, School of Basic Medical Sciences; Guangdong Provincial Key Laboratory of Single Cell Technology and Application, Southern Medical University, Guangzhou, Guangdong Province, China. [7]Department of Chemical and Biochemical Engineering, and School of Biomedical Engineering, The University of Western Ontario, London, ON, Canada. [8]Department of Applied Computer Science, University of Winnipeg, Winnipeg, MB, Canada. [9]State Key Laboratory of Oral Diseases & National Center for Stomatology & National Clinical Research Center for Oral Diseases, West China Hospital of Stomatology, Sichuan University, Chengdu, Sichuan, China. [10]These authors contributed equally: Renjie Qiu, Xingying Zhang, Chen Song. ✉e-mail: malcolm.xing@umanitoba.ca; wangleyu889@163.com

regarding the extensive application of CCPs: (1) the bioagent-free CCPs are inefficient in enhancing electroactivity improvement and restoring cardiac function, as current CCPs are solely considered as auxiliary carriers for cells and drugs[7], (2) the real-time feedback from CCP MI is desired, whereas it is a very high demand to integrate the electro-cardiosignal monitoring and effective CCP treatment.

Inspired by the electroactive property of cardiac tissue, electrical stimulation (ES) is an agreeable approach to induce the CMs' maturation[8,9], and even used to reduce the cardiac ischemic size after the ischemia-reperfusion injury elicited by an invasive electrode in the rat's ventricular wall[10]. Furthermore, the synergy of ES and conductive scaffolds can significantly improve cell-cell coupling and synchronous contraction of CMs, superior to the conductive scaffold itself[7]. Recently, avoiding the traditional battery-powered stimulation device, the complicated operation and the invasive implanted electrode, an innovative electrical generator called a triboelectric nanogenerator (TENG) has been deployed to cardiovascular system health care (Supplementary Data 1). TENG, serving as a green and infinite source of electricity, provides either the power supply for biomedical devices or for therapeutic electrodes that generate electrical stimuli. TENGs powered interdigital electrodes can promote the maturation of neonatal CMs, as well as increase and unify the beating rate of CMs[11,12]. Implantable TENGs (I-TENG) powered cardiac pacemakers successfully correct arrhythmia in large animal models[13–15]. On the other hand, the electrical output signals of TENGs, including open-circuit voltage, short-circuit current, and frequencies, are highly sensitive to mechanical motions and other stimuli, making it an excellent candidate for a miniaturized cardiac monitoring system. I-TENG have been thus transplanted into hearts for the detection of heart rate[16,17] and endocardial pressure[18]. Though I-TENGs hold great promise in the therapeutics and diagnosis of cardiac systems, I-TENGs that achieve MI repair and diagnosis simultaneously have yet to be developed. In addition, the therapeutic electrodes for TNEG-involved cardiac healing systems are typically constructed of inert metals such as gold. These electrode materials are stiffer than myocardium by several orders of magnitude, resulting in significant stiffness mismatch[19]. Moreover, sophisticated surface modification strategies are necessary to improve the effective contact area of the dielectric layers and the electrodes of TENG[20], which is not scalable. Accordingly, as patch treatment and diagnosis sensor for MI, a scalable miniature trinity (3 functions in 1 device) of TENG (TRI-TENG) CCP, encompassing the functions of CCP, self-power generation for non-invasive in situ electrical stimulation therapy, and real-time electrocardio monitoring, is called for but absent.

For MI treatment and diagnosis, in the design of our I-TENG CCP, a polydopamine (PDA) modified reduced graphene oxide (rGO) membrane is employed as a substitute for the metallic electrode. Our TRI-TENG CCP (TCP) adopts a unique double-spacer design with two spacers symmetric about a PDA-rGO membrane electrode. The first spacer sits on the myocardium, incorporating it as one component in the TRI-TENG. Thus, the PDA-rGO electrode raised by the first spacer works as a triboelectric electrode that generates triboelectric charges and simultaneously as the therapeutic electrode that builds an electric field on the myocardium, obviating the requirement for an additional therapeutic electrode (Fig. 1A). The second spacer, positioned on top of the PDA-rGO electrode, facilitates the contact and separation of the PDA-rGO with another triboelectric layer that exhibits higher ability to generate triboelectric charges. Owing to electrostatic induction, the electrical potential built between the myocardium and the PDA-rGO electrode is dictated by the higher electrical potential built between the PDA-rGO electrode and the triboelectric layer. We utilize mold casting to impart a leaf vein structure to bestow the polyvinylidene fluoride (PVDF) triboelectric layer with biomimetic leaf vein structure (Fig. 1A). The leaf vein structure and the PDA coating on the rGO electrode are both nature-inspired surface structures that can enhance the triboelectric effect by increasing the roughness and effective contact areas cost-effectively. The unity of the cardio patch, TENG-powered electrode, and sensor as three facets in one device is exhibited by our TCP. In detail, the TCP has three functions: (i) serves as the therapeutic electrode, which conveys electrical stimulus to infarcted tissues and facilitates electric signal transportation between normal and infarcted tissues, (ii) converts the biomechanical energy into electric energy, and (iii) serves as a potential wireless diagnosis device (Fig. 1B). With the combination property of conductivity and electrical generator, we hypothesize that the TCP produces a remarkable reparative effect on the infarcted heart in minipig MI models through the strengthened electroactivity reconstruction (Fig. 1C), surpassing the therapeutic efficacy of most recently reported approaches for MI treatment in minipigs (Supplementary Data 2).

## Results and discussion
### Assembly and characterization of TRI-TENG
As is illustrated in Fig. 1A, our TRI-TENG mainly comprised an elastomer bottom package, an rGO electrode, a PVDF triboelectric layer with leaf vein structure, Ecoflex 00-50 spacer, and a PDA-rGO electrode. The biocompatible elastomer Ecoflex 00-50 was used as a spacer and package to further augment the triboelectric effect and avoid leakage. According to literature reports, the weight loss of Ecoflex is less than 5% within 60 days and less than 10% within 200 days[21,22]. Ecoflex demonstrates minimal degradation over ~12 weeks, thereby ensuring the stability of TENG functionality. The TRI-TENG (8 mm in diameter, Supplementary Fig. 1A) underwent cyclic contact and separation with the contraction and relaxation of the heart, resulting in charges with opposite signs on the PDA-rGO electrode and the surface of the epicardium. Thus, the PDA-rGO patch membrane served a triple purpose, functioning as a conductive patch and as a triboelectric electrode for energy conversion, in addition to as a therapeutic electrode for the application of electrical stimuli to the epicardium. Meanwhile, the wireless sensing of the cardiac condition was achieved by connecting the rGO electrode to a Bluetooth-enabled device, facilitating communication with a smartphone application (Fig. 1B). To ensure compatibility with hearts possessing large surface areas, the TRI-TENG can be assembled in serial whose array design was employed to optimize the fitness and to a large scale (Supplementary Fig. 1B).

The bottom package as indicated by "1", the rGO electrode as indicated by "2", the leaf vein structured PVDF triboelectric layer as indicated by "3", and the PDA-rGO electrode as indicated by "4" are well demonstrated in the cross-section SEM image of our TRI-TENG (Fig. 1A). The bottom package was fabricated through the spin-coating and the following curing process. To prepare rGO electrodes and PDA-rGO electrodes, graphene oxide (GO) films were prepared in advance through drop casting GO aqueous solution on templates. During the evaporation process, GO sheets underwent self-assembly at the air/liquid surface and eventually formed a uniform GO film. The surface morphology of the formed GO film is shown in Supplementary Fig. 2 where each GO sheet can be identified clearly. The GO sheets were bumpy and loosely packed in the GO film, which is due to the distortion of the GO sheet caused by the presence of oxygen-containing functional groups (OCG)[23,24]. The energy dispersive spectroscopy (EDS) analysis was performed to obtain insight into the elemental composition. The C/O ratio of the GO film is 2.1 (Supplementary Fig. 3A). Fourier-transform infrared (FTIR) spectrum was used to study the OCG of the GO film (Supplementary Fig. 4A). The OCG-related peaks found in GO include the peak at 3143 cm$^{-1}$ corresponding to the stretching vibration of C−OH in the hydroxyl group, the peak at 1718 cm$^{-1}$ assigned to the stretching vibration of C=O in COOH, and the peak at 1030 cm$^{-1}$ attributed to the stretching vibration of C−O−C in epoxide. The peak at 1617 cm$^{-1}$ is due to in-plane vibrations of sp$^2$-hybridized C=C. These results were consistent with previous studies[25,26]. To prepare the rGO electrode, GO film was subjected to

thermal annealing at 300 °C. The rGO sheets were densely packed in the rGO electrode (Supplementary Fig. 2B), indicating the removal of OCG and restoration $sp^2$-hybridized lattice structure[27,28]. The C/O ratio of the rGO electrode increased to 4.2 (Supplementary Fig. 3B), which is another piece of evidence for the removal of OCGs. In addition, all the OCGs-related peaks disappeared in the FTIR spectrum of rGO, suggesting the successful reduction of GO. The sheet resistance of the rGO electrode was reduced from $756.93 \pm 44.471$ kΩ/sq to $0.420 \pm 0.047$ kΩ/sq (Supplementary Fig. 4B), further proving the reduction of GO. The rGO electrode was then subject to PDA coating deposition. After the treatment, PDA granules aggregated and anchored on the surface of the rGO electrode (Supplementary Fig. 2C). The PDA coating increased the atomic percent of oxygen and introduced nitrogen elements to the treated electrode (Supplementary

Fig. 3C). In the FTIR spectrum of the PDA-rGO electrode, the broad-band appeared at 3220 cm$^{-1}$, which was associated with the stretching vibration of N–H and O–H, and the two new peaks at 1508 cm$^{-1}$ and 1050 cm$^{-1}$, which were associated with the stretching vibration of C = N and C–N[29], further proving the successful anchoring of polydopamine. The sheet resistance of the PDA-rGO electrode ($0.885 \pm 0.036$ kΩ/sq) was slightly larger than the rGO electrode (Supplementary Fig. 4B). PVDF dissolved in DMF/Acetone was drop cast on a Ecoflex mold to prepare the PVDF layer with leaf vein structure. The surface of the as-prepared PVDF layer processed delicate leaf vein structure as is shown in Supplementary Fig. 2D.

We detected the cell viability of neonatal rat CMs cultured on different substrates. As shown in Supplementary Fig. 5, cell viabilities of CMs cultured on the non-conductive Ecoflex, a conductive substrate

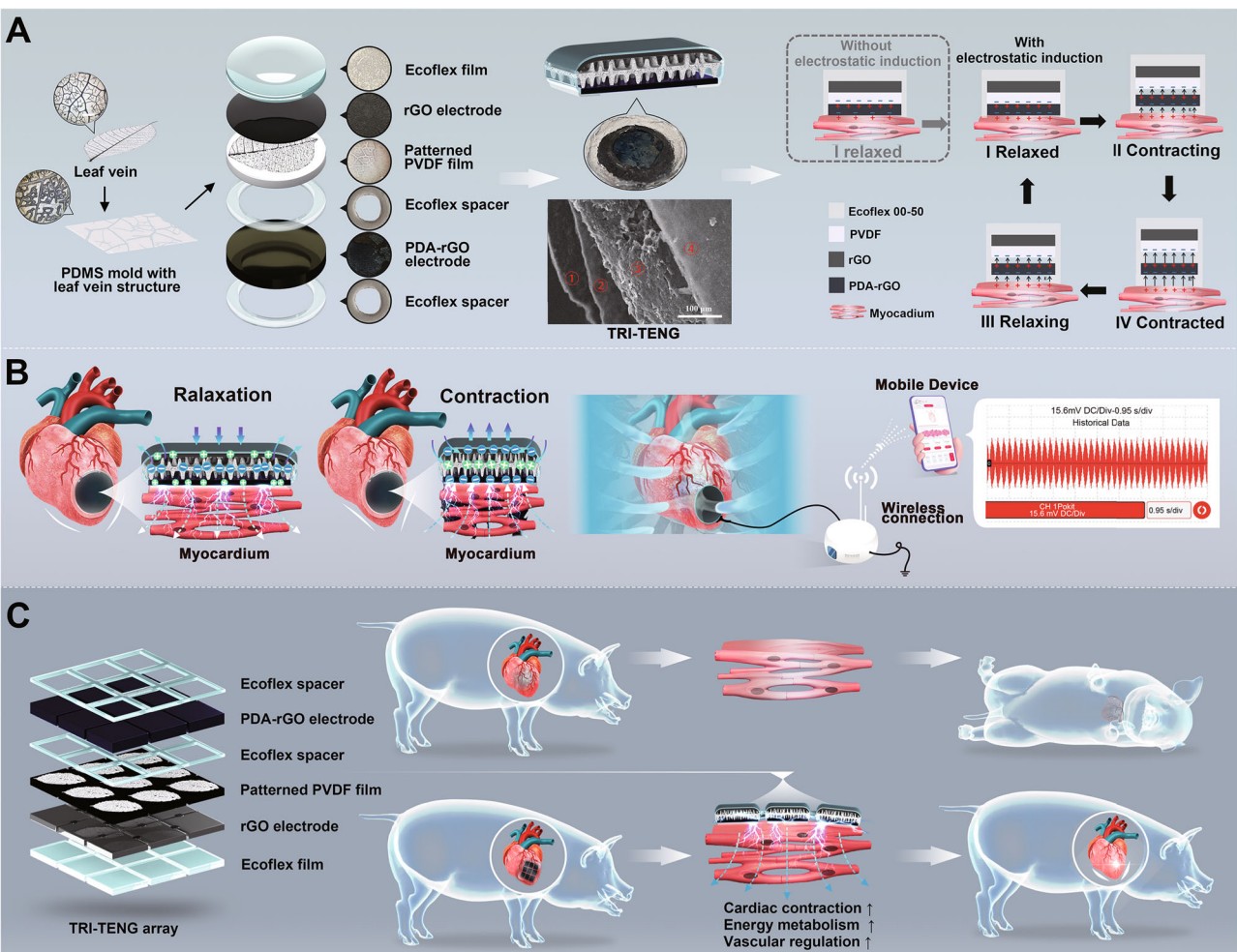

**Fig. 1 | The preparation, characterization, working mechanism, and potential application of TRI-TENG. A** The first column shows the schematic illustration of the preparation of the polyvinylidene fluoride (PVDF) triboelectric layer with leaf vein structure using mold casting and detailed composition of trinity triboelectric nanogenerator (TRI-TENG). The second column shows the schematic diagram (upper), actual photograph (middle) and scanning electron microscopy (SEM) image of the TRI-TENG on its cross-section (lower) showing multilayered structures: ①, Ecoflex film, ②, reduced graphene oxide (rGO) electrode, ③, polyvinylidene fluoride (PVDF) film with leaf vein structure, ④, polydopamine (PDA) modified rGO (PDA-rGO) electrode. The third column shows the healing mechanism that an electrical potential built between the PDA-rGO electrode and the myocardium upon the contraction and relaxation of the myocardium. The as-built electrical potential is equal to the electrical potential between the leaf vein patterned PVDF layer and the PDA-rGO electrode due to electric induction. **B** An electric system that acquires the open-circuit voltage ($V_{OC}$) between the rGO electrode and the ground

resulting from heart activity and transmits it to a smartphone wirelessly. **C** Schematic of TRI-TENG array assembly for matching the infarct size of the porcine heart and its application in a minipig MI model. The preparation of each layer structure of TRI-TENG array adopts the same configuration as TRI-TENG using the polylactic acid (PLA) mold, and every two neighboring rGO electrodes as well as PDA-rGO electrodes were electrically connected by an air-dried poly(3,4-ethyle-nedioxythiophene):poly(styrenesulfonate)/methacrylated gelatin (PEDOT:PSS/GelMA) hydrogel. After 28 days of transplanting the TRI-TENG array into the infarcted heart, there was a significant improvement in cardiac function by approximately 14.7%, surpassing the therapeutic efficacy of most recently reported approaches for MI treatment in minipigs. We hypothesize that TRI-TENG conductive cardiac patches (CCPs) exert their therapeutic effects on infarcted hearts primarily by modulating the expression of mRNA related to cardiac muscle contraction, energy metabolism, and vascular regulation in vivo, as revealed by RNA sequencing analysis.

of PDA-rGO electrode in the PDA-rGO/Ecoflex (PDE) group and on the self-powered triboelectric substrate (TRI-TENG) were very well during 7 days culture. The CMs' survival rate in the PDE group and the TRI-TENG group were higher than 80% on days 3 and 7 of culture. The biocompatibility of these two biomaterials is similar to that of other existing implantable self-powered materials (Supplementary Table 1). In addition, the Young's modulus of the TRI-TENG was calculated to be 640.70 ± 71.07 Kpa, aligning with most the scaffolds used in MI management (Supplementary Fig. 6). These results suggest that the composition of TRI-TENG exhibits good biocompatibility, and its mechanical properties are well-suited for CMs and cardiac tissues.

### The performance of TRI-TENG as the energy convertor in vitro and in vivo

To harvest the mechanical energy produced by heart contraction and relaxation for electrical stimuli generation, the TRI-TENG adopted a contact-separation work mode. The working mechanism of the TRI-TENG is depicted in Fig. 1A. At the initial state when the heart was contracted, there was no contact between the PDA-rGO electrode and the PVDF layer because of the existence of the Ecoflex spacer. As the heart started to relax, the TRI-TENG was stretched out, forcing simultaneous contact between the upper surface of the PDA-rGO electrode and the PVDF layer, as well as between the lower surface of the PDA-rGO electrode and the epicardium. According to Supplementary Fig. 7, the PVDF layer tends to gain electrons compared with the PDA-rGO electrode, and the PDA-rGO electrode tends to gain electrons compared with the myocardium. Thus, electrons can transfer from the upper surface of the PDA-rGO electrode to the PVDF layer during the contact, leaving positive charges on the upper surface the PDA-rGO electrode. Electrons are also injected from the myocardium to the bottom layer of the PDA-rGO electrode, resulting in negative charges at the bottom surface of the PDA-rGO electrode and positive charges in the epicardium. As a result of electrostatic induction, triboelectric charges generated on the upper surface of the PDA-rGO electrode were equal to those generated on the bottom surface of the PDA-rGO electrode. The three layers were in immediate contact when the heart was fully relaxed. There was barely any distance between charges with opposite signs, resulting in no potential difference between any two oppositely charged surfaces. As the heart started to contract, the distance between two oppositely charged surfaces gradually increased, and the electric potentials between the PDA-rGO electrode and the epicardium as well as between the PDA-rGO electrode and the PVDF layer started to establish. The potential differences reached the maximum value when the heart was fully contracted. The first spacer enables the PDA-rGO electrode to not only takes part in the generation of triboelectric charges but also work as a therapeutic electrode that builds an electric field on the myocardium. The second spacer amplifies the built electric field to the same strength as the one built between the PDA-rGO and the PVDF.

To function as a sensor, the output voltage between the rGO electrode of the TCP and the ground was tested by an electrometer or wireless sensing module (Supplementary Fig. 8). To balance the gradually increasing potential difference built between the top surface of the PDA-rGO electrode and the PVDF layer, across the second spacer, during heart contraction positive charges would be driven to flow from the ground to the rGO electrode. The flow of the positive charges ends when the heart is completely contracted. Once the layers were forced to approach each other during heart relaxation, the positive charges would be repelled back to the ground. Thus, heart contraction and relaxation lead to the generation of the alternative current voltage between the rGO electrode and the ground. Moreover, the electrical potential built between the bottom surface of the PDA-rGO electrode and the myocardium, across the first spacer, remains unaffected by the measuring event involving the second spacer. When the rGO electrode is connected to a wireless module, the TCP can operate as a wireless sensor, self-powered electrical stimulus generator, and conductor simultaneously. We designed a model study to prove that the sensing unit and stimulation do not interfere with each other, regardless of the complexity of in vivo voltage measurement. As shown in Supplementary Fig. 9 and Supplementary Movie 1, the field potential on the sodium alginate (SA) hydrogel and the voltage output from the rGO electrode can be simultaneously measured by the electrocardiography (ECG) electrode and the multimeter, when the SA hydrogel is subject to cyclic compression. Moreover, the peak value of both the field potential and the voltage output can remain stable. Thus, the sensing unit and the stimulation unit don't interfere with each other, without consideration of the complexity of measuring voltage output in vivo.

Nature-inspired surface structures were employed to improve the electrical output performance of TCP. An electrometer was connected between the PDA-rGO electrode and the rGO electrode to test the performance of different TENGs with different components at controlled pressure and loading rate. The leaf vein structure on the Patterned PVDF layer (P-PVDF) resulted in significant improvement in the open-circuit voltage, short-circuit current, and transferred charges (Fig. 2A). As shown in Supplementary Fig. 2C and Supplementary Fig. 10, the PDA coating aggregated on rGO surface as granules, which significantly increases the roughness of PDA-rGO electrode. The charge density of the generated triboelectric charges on the surface of the PDA-rGO electrode is higher than that on the rGO electrode. Consequently, the PDA coating on the PDA-rGO electrode further increased the open-circuit voltage, short-circuit current, and transferred charges to 21.98 mV, 2.23 nA, and 0.22 nC, respectively. To investigate the output power of TENGs, resistors with resistance spanning from $1\,K\Omega$ to $1\,G\Omega$ were connected as external loads. The output voltages and currents of all the TENGs remained stable when the external load was less than $1\,M\Omega$ (Fig. 2B). When the external load exceeded $1\,M\Omega$, the voltages of all TENGs dramatically rose with the increase in the resistance of the external load, while the currents of all TENGs dropped noticeably due to Ohmic loss. Consequently, the instantaneous output powers of all the TENGs reached their maximum values at $10\,M\Omega$. The instantaneous output power of TENGs was also increased by the nature-inspired leaf vein structure and the PDA structure. The TENG with both the P-PVDF and the PDA-rGO electrode reached the highest maximum instantaneous output power of $0.16\,\mu W/m^2$.

The contraction and relaxation of the heart occasionally exhibit irregularity in terms of strength and frequency. The amplitude, frequency, and waveform of the TENG output signals are highly dependent on external mechanical stimuli, a phenomenon that has been reported in other TENG sensors[30]. Thus, the voltage output of the TCP was evaluated under different strains, compression frequencies, and different compression strengths. The amplitude of the output voltage increased as the compression frequency increased from 1 to 5 Hz (Fig. 2C). In addition, the frequency of the output voltage can serve as a representative measure of the frequency of the applied compression. The amplitude of the output voltage also increased with the increase in the applied pressure. The relationship between the relative change in the amplitude output voltage and pressure is demonstrated in Supplementary Fig. 11A, the response of the amplitude to pressure can be divided into two regions, namely the high-sensitivity Region I and the low-sensitivity Region II. In Region I where the pressure is less than 5.9 kPa, the sensitivity of the TRI-TENG sensor was 6.74 mV/kPa ($R^2 = 0.979$). The sensitivity of the TRI-TENG sensor dropped to 2.54 mV/kPa ($R^2 = 0.991$) in the high-pressure region (Region II). The voltage amplitude of the TRI-TENG sensor depends on the change in the spatial distance between the P-PVDF layer and the PDA-rGO electrode, the speed of change in the spatial distance, and triboelectric charge density[31]. In Region I, pressure increase caused a substantial increase in the change of spatial distance, as a result, voltage amplitude changed dramatically. At the turning point between Region I and Region II, the spatial distance was probably close to zero, and the

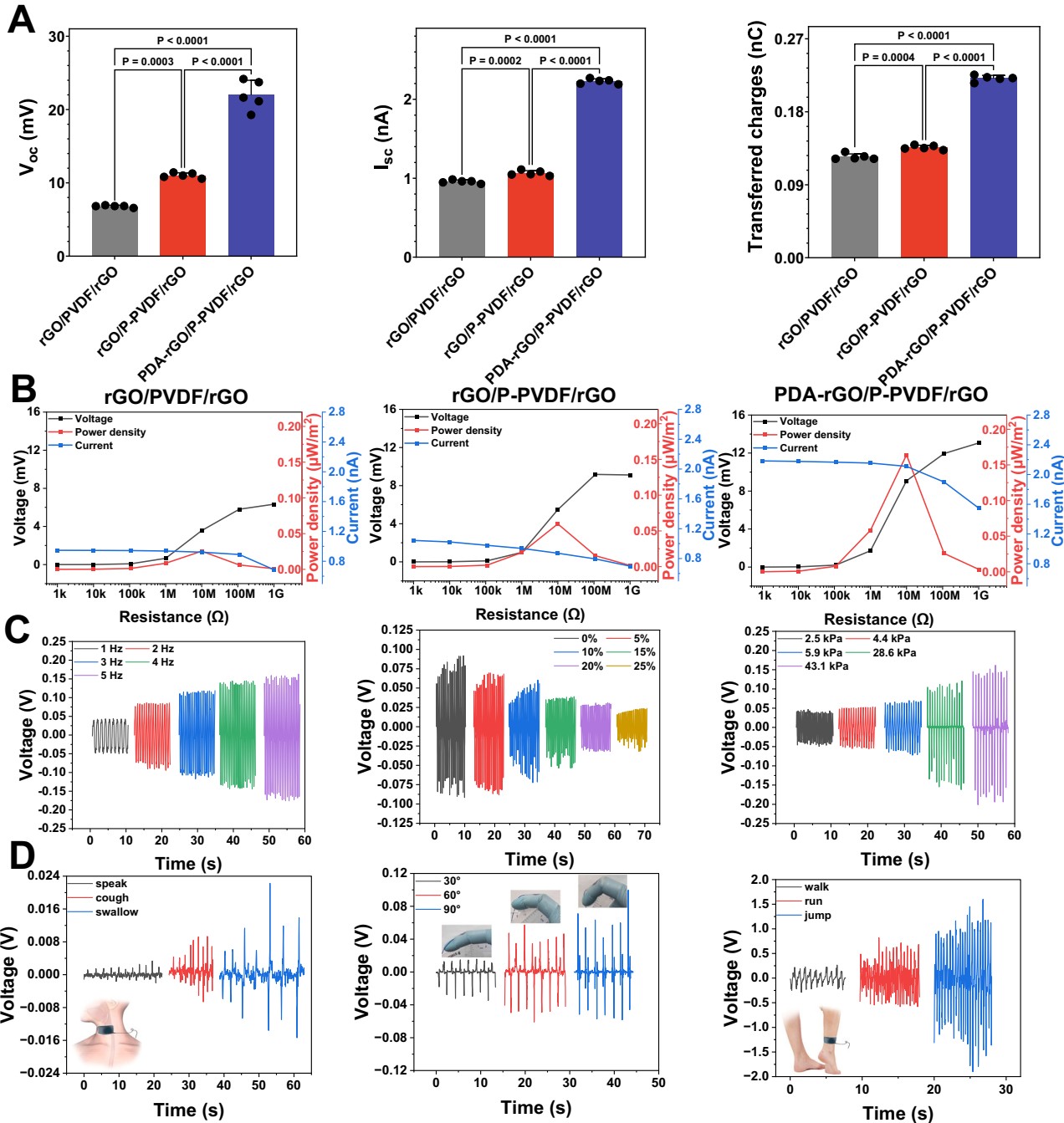

**Fig. 2 | The in vitro output performance of different TENGs and the response of TRI-TENG to different stimuli. A** The performance of different TENGs with different components in the open-circuit voltage ($V_{OC}$), short-circuit current ($I_{SC}$), and transferred charges ($n = 5$ independent samples). **B** Output voltage, power density, and current of different TENGs at different load resistance. **C** $V_{OC}$ of TRI-TENG at different frequencies, pressure, and strains. **D** Demonstration of the TRI-TENG detecting different mechanical motions. The TRI-TENG is attached to the throat for detecting speaking, coughing, and swallowing; attached to the fingers for detecting finger joint movements; and attached to the ankle for detecting walking, running, and jumping. Statistical significance in sample of $V_{OC}$ was calculated using two-side one-way ANOVA with Dunnett's post hoc test, and statistical significance in sample of ISC and transferred charges was calculated using two-side one-way ANOVA with LSD post hoc test.

increase in pressure might merely increase the contact area between the two layers, exerting little effect on the output voltage. The output voltage of the TRI-TENG sensor decreased with the increase in the applied strain (Supplementary Fig. 11B). Due to the positive Poisson's ratio of the elastomer spacer, the spatial distance between the two layers decreases proportionally with the increasing strain. The output voltage of the TRI-TENG sensor exhibited a decrease in response to the applied strain, with a sensitivity of $-2.65 \, \text{mV}/1\%$ ($R^2 = 0.970$) (Supplementary Fig. 11B).

A single variation in the strain or pressure of mechanical stimuli results in a change in the output voltage, and the frequency of the output voltage represents the frequency of mechanical stimuli. The TCP's potential as an activity monitoring sensor was assessed. The TCP sensor was attached to different sites of the human body by a tape for the measurement of different human activities. The motion of the throat involves small-strain vibrations occurring at a specific frequency. Thus, a particular movement of the throat produced a distinct change in spatial distance, resulting in a unique waveform of output

voltage (Fig. 2D and Supplementary Movie 2). Large-strain mechanical deformation such as the bending index finger can also be similarly monitored by the TCP sensor (Fig. 2D and Supplementary Movie 3). The peak output voltage of the TENG increased with the increasing bending angle. The increment in bending angle from 30° to 60° resulted in a more significant increase in peak output voltage compared to the increase from 60° to 90°. The change in the spatial distance may reach saturation after the bending angle exceeds 60°. Despite variation in bending angles, finger bending followed the same movement regime. Thus, the output voltage exhibited a consistent waveform as the bending angle increased. The TCP fixed on the ankle can measure the level of human activity (Fig. 2D and Supplementary Movie 4). As the intensity of human activity increased, there was an increase in both the amplitude and frequency of the output voltage. To further investigate the stability and repeatability of TCP under long periods of cycles, we tested 1000 cycles. As shown in Supplementary Fig. 12, the amplitude of the output voltage remains relatively stable even after a large number of compression and relaxation cycles. These evidences showed that our TCP still has good performance under long periodic action, which is suitable for the periodic movement of the heart.

In order to evaluate the performance of TRI-TENG as the sensing device in vivo, TRI-TENG array was transplanted between the apex cordis and pericardium in the minipig, and the TRI-TENG's electrical output and the ECG signals were simultaneously recorded (Fig. 3A, B). As shown in Fig. 3C, the fluctuating frequency of the open-circuit voltage ($V_{OC}$) outputted from the TRI-TENG array was fully consistent with the heartbeat frequency derived from ECG, and the R–R intervals in ECG were equivalent to the peak-peak phases in $V_{OC}$ wave. This result indicates that the voltage generated by the TRI-TENG is triggered by cardiac movement. The periodical contraction and relaxation of the heart prompt the friction layer of TRI-TENG, resulting in contact and separation, which finally produces an electrical output through the TRI-TENG. Driven by the porcine beating activities, the mean maximum open-circuit voltage ($V_{OC, max}$) and the mean minimum open-circuit voltage ($V_{OC, min}$) produced from the TRI-TENG's output signals were $5.975 \pm 1.438$ mV and $-6.512 \pm 1.665$ mV (Fig. 3C, H). Furthermore, the TRI-TENG was transplanted onto the Langendorff-perfused isolated rat heart to observe directly the corresponding $V_{OC}$ changes outputted from TRI-TENG along with the change of the heart's contraction force (Fig. 3D, E and Supplementary Fig. 13A, B). In sinus rhythm, the mean $V_{OC,max}$, $V_{OC,min}$, and the open-circuit voltage difference ($\Delta V_{OC}$) from the Langendorff-perfused rat heart were $0.664 \pm 0.109$ mV, $-0.546 \pm 0.140$ mV, $1.210 \pm 0.233$ mV respectively, which were just in the range of the non-excitatory electrical stimulation applied on rat heart (0.1 - 1 mV) (Supplementary Fig. 13A)[10]. When the heart rate was elevated to 420 b.p.m by 7 Hz pacing, all the $V_{OC,max}$, $V_{OC,min}$, and $\Delta V_{OC}$ produce by TRI-TENG were increased correspondingly compared with those in sinus rhythm (Supplementary Fig. 13A). However, when the left ventricle (LV) was ischemic injured by the ligation of the left anterior descending coronary artery, the related $V_{OC}$ values were decreased whether in the sinus rhythm or the loaded 7 Hz pulse (Supplementary Fig. 13B). In addition, we investigated the effect of reduced cardiac contractile motion in the ischemic region on the TRI-TENG' function. As shown in Supplementary Fig. 14, a significant improvement in field potential amplitudes of the heart after the TRI-TENG transplantation under the ischemic conditions and the $\Delta V_{OC}$ values of the TRI-TENG driven by ischemic myocardium can be detected, indicating that the reduced cardiac contractile motion can also induce electrical energy generation by the TRI-TENG.

Wireless transmission is essential for implantable sensors for post-treatment monitoring. Given that TRI-TENG can sensitively detect the voltage changes of normal and ischemic hearts, we utilized an external device (Pokit meter) to detect and wirelessly transmit the in vivo output voltage signal from TRI-TENGs driven by rat hearts under

different states, and the wirelessly transmitted signals were received by a mobile phone for real-time analysis (Fig. 3F and Supplementary Movie 5). The output signals obtained from wireless transmission displayed the same variation trend with those by wired measurement, and the output voltage values were significantly decreased under the ischemic heart state compared with those under the normal heart state (Fig. 3G, I, J and Supplementary Fig. 13C). We also investigated the influence of breathing movement on the sensing function of the TRI-TENG. As shown in Supplementary Fig. 15, the electrical output capacity of the TRI-TENG is stable under both open-chest and closed-chest conditions. Collectively, these results suggest that this TRI-TENG was an effective energy convertor, which can output the synchronized electrical signals with the ECG by harvesting the heart's biomechanical energy. The cardiac contraction and relaxation activities result in the periodic separation and contact of the two triboelectric layers in TRI-TENG[16], which made it acts as an heart activity monitoring sensor. In addition, this TRI-TENG was sensitive to the changes of cardiac contractility. When myocardial ischemia, contractile force becomes weak, leading to a significant decrease in the output voltage value of the TRI-TENG, which allows for precise monitoring of cardiac signals under pathological state. Here, TRI-TENG can also connect to wireless devices and transmit signals to the mobile phone to enable real-time monitoring of cardiac electrical signals, which holds significant implications for the timely diagnosis of pathological cardiac conditions.

## Effects of TRI-TENG on CM structure and maturation

Our previous studies suggest that the surface topography and conductivity of the cardiac patches can remodel the cell phenotype and function of CMs in vitro, which may be an important element for CPs to activate endogenous repair after transplantations. In this study, the cell shape and the cardiac-specific proteins expression of neonatal rat CMs cultured on Ecoflex, PDE and TRI-TENG were detected, respectively. On day 3 of culture, more CMs and a larger spreading area of CMs were observed in the PDE and TRI-TENG groups through F-actin staining, compared with those in the Ecoflex group (Supplementary Fig. 16A, D, E). On day 7 of culture, more dense and elongated myofibrils dotted with massive parallel aligned actin, which were closely related to the CMs' differentiation and maturation, were detected in PDE and TRI-TENG compared with those in the Ecoflex group (Supplementary Fig. 16A, D, E). SEM images also showed that confluent CMs-formed myocardial-like structures were on the PDE and TRI-TENG, and the direct contact and intercellular communication were clearly visible (Supplementary Fig. 16B). Immunostaining of cardiac-specific markers, sarcomeric α-actinin, and CX43, in CMs in different groups displayed that more mature sarcomeric structures and CX43 expressions were located in PDE and TRI-TENG compared those in Ecoflex on days 3 and 7 of culture (Supplementary Fig. 16C). Quantificational analyses results showed that the highest α-actinin and CX43 coverage area were presented in the TRI-TENG group (Supplementary Fig. 16F, G). CX43, the main component of gap junction, plays a vital role in transmitting the electrical excitation signals among CMs[32]. Accordingly, these results indicate that the PDE CCP benefits CMs' maturation and synchronous electrical excitation. PDA-rGO had excellent properties of water-stability and biocompatibility, thus, the PDA-rGO electrode absorbed more proteins through hydrophobic interactions, electrostatic attraction or π-π stacking, leading to high CM density on it[33]. In addition, PDA-rGO can reduce CMs' excitation threshold and accelerate the electrical signal transmission among CMs[34]. Furthermore, the additional electrical stimulation in TRI-TENG can facilitate fast maturation and functionalization of CMs.

## TRI-TENG enhances the electroactivity of the infarcted heart in rat MI models

The effects of TRI-TENG on the electrical properties of the injured myocardium were further studied. The infarcted myocardium had an

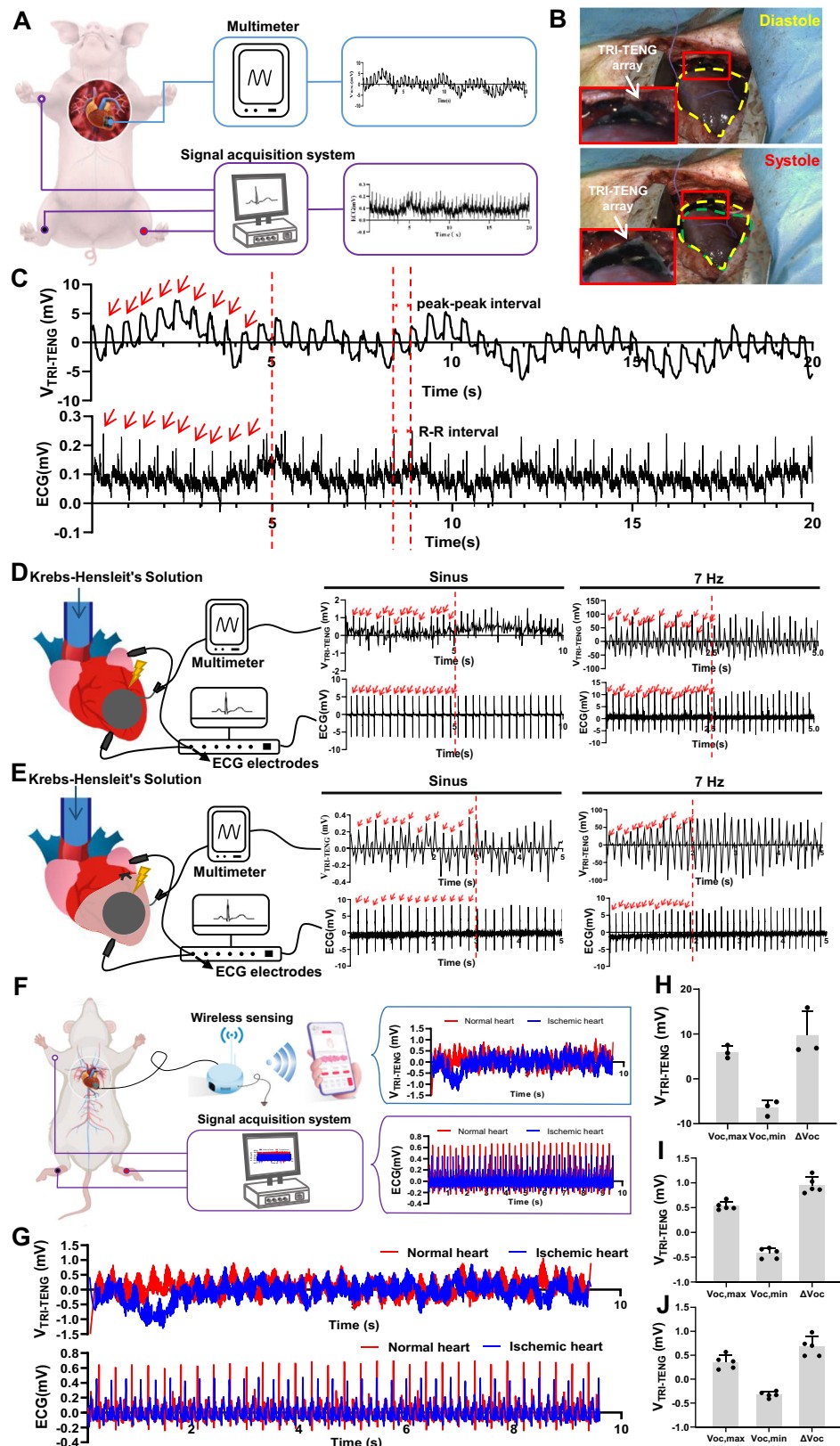

inefficient movement ability because of the uncoupling of excitation–contraction, leading to decompensated hypertrophy and deteriorated cardiac function[35]. The enhanced electrical sensitivity and contractility enable the injured myocardium to regain movement vitality to avoid cardiac function deterioration[36]. As illuminated in Fig. 4A–C, the ligation of the left anterior descending coronary artery

and the injured myocardium deadened the electrical sensitivity of the Langendorff-perfused rat heart, demonstrated by the increased pacing thresholds. Interestingly, the TRI-TENG's transplantation reduced the pacing thresholds of rat heart, that is, lower stimulus voltage pulses were enough to inspire the whole hearts' synchronous pacing either on the TRI-TENG-transplanted normal heart or on the

**Fig. 3 | The application potential of the TRI-TENG as a real-time electrocardio sensor in vivo. A** Schematic of the electrical signals outputed from TRI-TENG in swine and the electrocardiography (ECG) signals recorded simultaneously by the signal acquisition system. **B** Representative macroscopic images of the TRI-TENG array placed between the apex cordis and pericardium in minipig, which was driven by the beating activities of the heart. The yellow line represents the diastolic cardiac contour and the green line represents the systolic cardiac contour. **C** $V_{OC}$ from TRI-TENG and 2-lead ECG in minipigs. The upper wave represents $V_{OC}$ and the lower wave represents ECG. The number of marked peaks in $V_{OC}$ was in accordance with the marked QRS peaks in ECG from the same 5-s period and the marked peak-peak interval in $V_{OC}$ was matched the R–R interval in ECG, which showed the correlation between voltage and ECG signals. **D, E** The $V_{OC}$ and the ECG were simultaneously recorded from the TRI-TENG-transplanted Langendorff-perfused rat normal heart (**D**) and ischemic injured heart (**E**). The signals were displayed under sinus rhythm and 7 Hz stimulation pacing, respectively. The number of marked peaks was consistent in both the sinus rhythm and 7 Hz-stimulated heart. **F** Schematic diagram illustrating the electrical output assessment of TRI-TENG in rats as a self-sustaining wireless sensor. **G** Simultaneously recording of $V_{OC}$ and ECG from a TRI-TENG -transplanted rat heart under normal and ischemic states. **H–J** Statistics analyses of maximum open-circuit voltage ($V_{OC, max}$), minimum open-circuit voltage ($V_{OC, min}$), and the open-circuit voltage difference ($\Delta V_{OC}$) respectively from the minipig hearts ($n = 3$ independent experiments) (**H**), the in vivo hearts under the normal state ($n = 5$ independent experiments) (**I**) and under the ischemic state ($n = 5$ independent experiments) (**J**). The data were presented as mean ± SD.

TRI-TENG-transplanted injured heart (Fig. 4B, C and Supplementary Tables 2 and 3). Next, we investigated whether the TRI-TENG's enhanced electrical sensitivity in the heart by the transplantation is coupled with the heart's contractility (Fig. 4D–I). Stimulated by the same electrical pulse, LV contractility was elevated about 2 times once the TRI-TENG was transplanted onto the rat LV compared with that before transplantation (Fig. 4F, G). After TRI-TENG's transplantation on the infarcted heart in rat MI models for 4 weeks, the contractility of the infarcted LV was significantly increased compared with that in the MI group (Fig. 4H, I). With the weak conductive property of Ecoflex, its application on the normal heart or the injured heart had no influence on the heart's pacing thresholds and contractility (Fig. 4D–I and Supplementary Tables 2 and 3). As for the conductive PDE patch, its transplantation had no obvious effect on the electrical sensitivity of the normal heart or the ischemic heart and had no obvious effect on the contractility of the normal myocardium (Fig. 4D–I and Supplementary Tables 2 and 3). After being transplanted on the rat's infarcted heart for 4 weeks, the PDE patch-treated LV contractility was higher than that in the MI group (Fig. 4H, I). These results indicated that the microcurrent produced by TRI-TENG had an instant and sustained effect on the electrical excitation for the injured heart, which can be further coupled to the contraction of the myocardium, leading to the increased contractility of the infarcted heart. The conductive patch, however, seemed to have a minimal compact on the instant electrical excitation and contractility of the heart. It was supposed that the conductive patch transplantation elevated the contractility of the infarcted heart through an indirect way, that is, the structural and functional recovery of the infarcted heart elicited by the conductive patch's transplantation for 4 weeks results in the contractility promotion[37].

In addition, electrical mapping and optical mapping were also performed to probe the electrophysiology reconstruction of the infarcted heart after the patches' transplantation for 4 weeks. Epicardial electrical mapping, through collecting and analyzing the electrical activity of the LV free wall, generated electrical conduction velocity (CV) maps from the non-infarcted myocardium to the infarcted myocardium[38]. As shown in Fig. 4F, the maps of the MI group and the Ecoflex group were characterized by inhomogeneous conduction, and their electrical propagation was delayed. The PDE or the TRI-TENG transplantation significantly accelerated the excitation propagation between healthy and infarcted myocardium, and the TRI-TENG transplantation achieved the highest CV among those transplantation groups (Fig. 5A, D). The surface ECG traces in different groups were simultaneously recorded with the electrical mapping operation (Fig. 5B). In agreement to the CV maps' exhibition, the broadened (prolonged) QRS durations in ECG recordings reflected the abnormal ventricle conduction in the MI and the Ecoflex groups (Fig. 5B, E)[39]. The PDE or TRI-TENG transplantation recovered QRS durations, and it is inspiring that the produced QRS durations by the TRI-TENG transplantation were close to that in the sham group (Fig. 5B, E). The field potentials in remote, border and scar regions of the heart in different groups were also measured by the electrical mapping system. According to Fig. 5C, the Ecoflex transplantation had no obvious effect on the global field potential of the infarcted heart. Both the border/remote field potential amplitude and the scar/remote field potential amplitude ratios in the PDE-transplanted heart were increased compared with that in MI heart, though the difference of the border/remote field potential amplitude ratio between the PDE group and the MI group had no significance (Fig. 5C, F, G). The TRI-TENG-treated infarcted hearts had the highest border/remote field potential amplitude ratio and the scar/remote field potential amplitude ratio among the three transplantation groups of hearts (Fig. 5C, F, G). These analytical results suggest that after being transplanted for 4 weeks, the conductivity and the conductivity plus self-powered nanogenerator can improve the regional electrical activity and strengthen the global electrical impulse propagation of the infarcted heart. The TRI-TENG transplantation achieved the optimum effects on them.

Subsequently, optical mapping was conducted to assess the transmembrane action potential ($V_m$) and the $Ca^{2+}$ holding dynamics in Langendorff-perfused isolated hearts in different groups. The electrical propagation and calcium transient fluorescent signals were captured from the RH237 (a voltage-sensitive dye) and Rhod-2 AM (a $Ca^{2+}$ dye)-staining hearts during a 5 Hz point stimulation (Fig. 5H), respectively. As predicted, reflected by the representative maps of action potential (AP) propagation and $Ca^{2+}$ activity, MI perturbed the AP propagation and caused $Ca^{2+}$ mishandling in the left ventricular myocardium (Fig. 5I, J). However, TRI-TENG transplantation greatly benefits normalizing AP propagation and $Ca^{2+}$ dynamics in the infarcted left ventricle (Fig. 5I, J). The disordered electrical propagation and calcium transient patterns usually result in repolarization variations in myocardium[4]. As indicated in Fig. 5K, L, MI prolonged action potential duration at 90% repolarization ($APD_{90}$) and calcium transient durations at 90% recovery ($CaD_{90}$). After TRI-TENG treatment on the infarcted heart for 4 weeks, the disorganized electrophysiological function of the infarcted heart recovered, and the prolonged $APD_{90}$ and $CaD_{90}$ were significantly decreased. Accordingly, these results suggest that TRI-TENG transplantation can accelerate the repolarization of the injured ventricular myocardium, and reduce the risk of calcium-dependent gap junction uncoupling and the malignant arrhythmia in the infarcted heart.

## TRI-TENG therapy for infarcted heart in rat and porcine MI models

The commendable performance of TCP on enhancing electroactivity suggests a promising reparative effect for injured myocardial tissue. Accordingly, TCP's repair effect for infarcted heart was detected in rat and minipig MI models at 4 weeks post-transplantation. We firstly verified that there was little effect on normal heart structure and function at 4 weeks after TRI-TENG transplantation (Supplementary Fig. 17). Masson's trichrome staining for cardiac sections in rat experiments exhibited that the infarcted myocardium was almost completely replaced by scar tissue in the MI group, along with the most

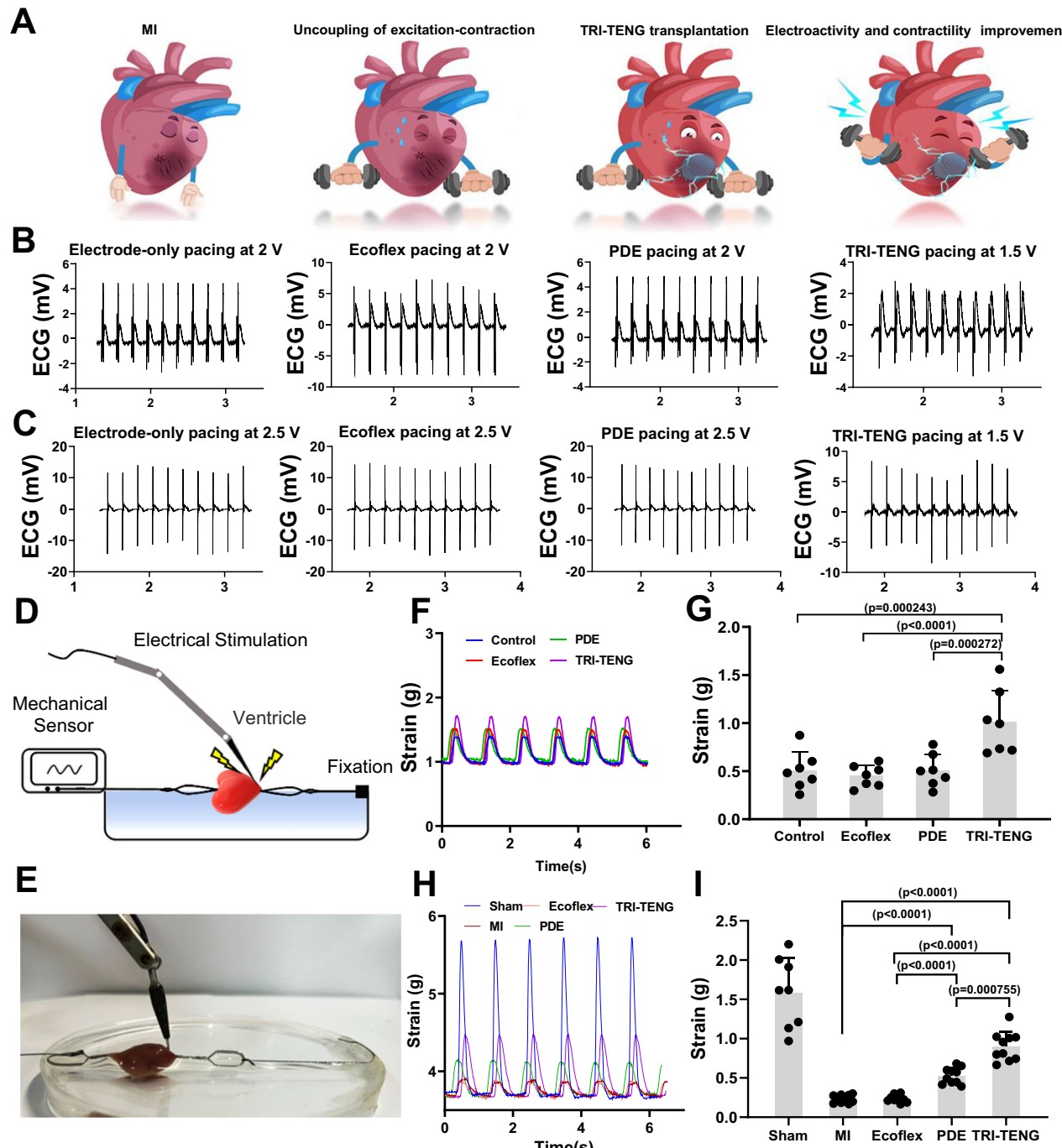

**Fig. 4 | The impact of TRI-TENG transplantations on the excitation–contraction coupling of rat hearts. A** Schematic illustrating the augmented effects of TRI-TENG transplantation on excitation–contraction coupling in the infarcted rat heart. **B, C** Pacing thresholds of the Langendorff-perfused whole heart in different groups under various normal condition (**B**) and ischemic condition (**C**). **D, E** Schematic representation (**D**) and Profile display (**E**) of contractility measurements in ventricular tissue. **F–I** Contraction waves (**F, H**) and contractility analyses (**G, I**) of the ventricular tissues were assessed under 1 Hz pulse stimulation in different groups following the patches' transient transplantation on the normal heart (**F, G**) or patches' transplantation on the infarcted heart for a duration of 4 weeks (**H, I**). Each symbol denotes an independent rat (**G**: *n* = 7 in each group. **I** Sham *n* = 8, and *n* = 10 in the other groups). The data were presented as mean ± SD. Statistical significance in (**G**) was calculated using two-sided one-way ANOVA with LSD post hoc test, and statistical significance in (**I**) was calculated using two-sided one-way ANOVA with Dunnett's post hoc test.

extensive infarct area and the thinnest left ventricular wall thickness. The Ecoflex group exhibited negligible differences from the MI group in terms of infarct size and LV wall thickness, while the PDE and TRI-TENG groups demonstrated a pronounced effect on reducing fibrosis in the infarct region. Specifically, these two groups showed significantly decreased infarct size and increased LV wall thickness (Fig. 6A–C). Consistent with Masson's staining results, significant

expression of α-actinin and CX43 proteins was observed in the infarct region of the hearts in the PDE and TRI-TENG groups, while minimal expression was detected in the MI and Ecoflex groups (Fig. 6D and Supplementary Fig. 18B, C). Furthermore, dual immunostaining for vWF and α-SMA proteins in the infarct region from different groups revealed that the PDE or the TRI-TENG transplantation elevated vWF⁺ microvessels and vWF⁺/α-SMA⁺ arterioles in the infarct regions,

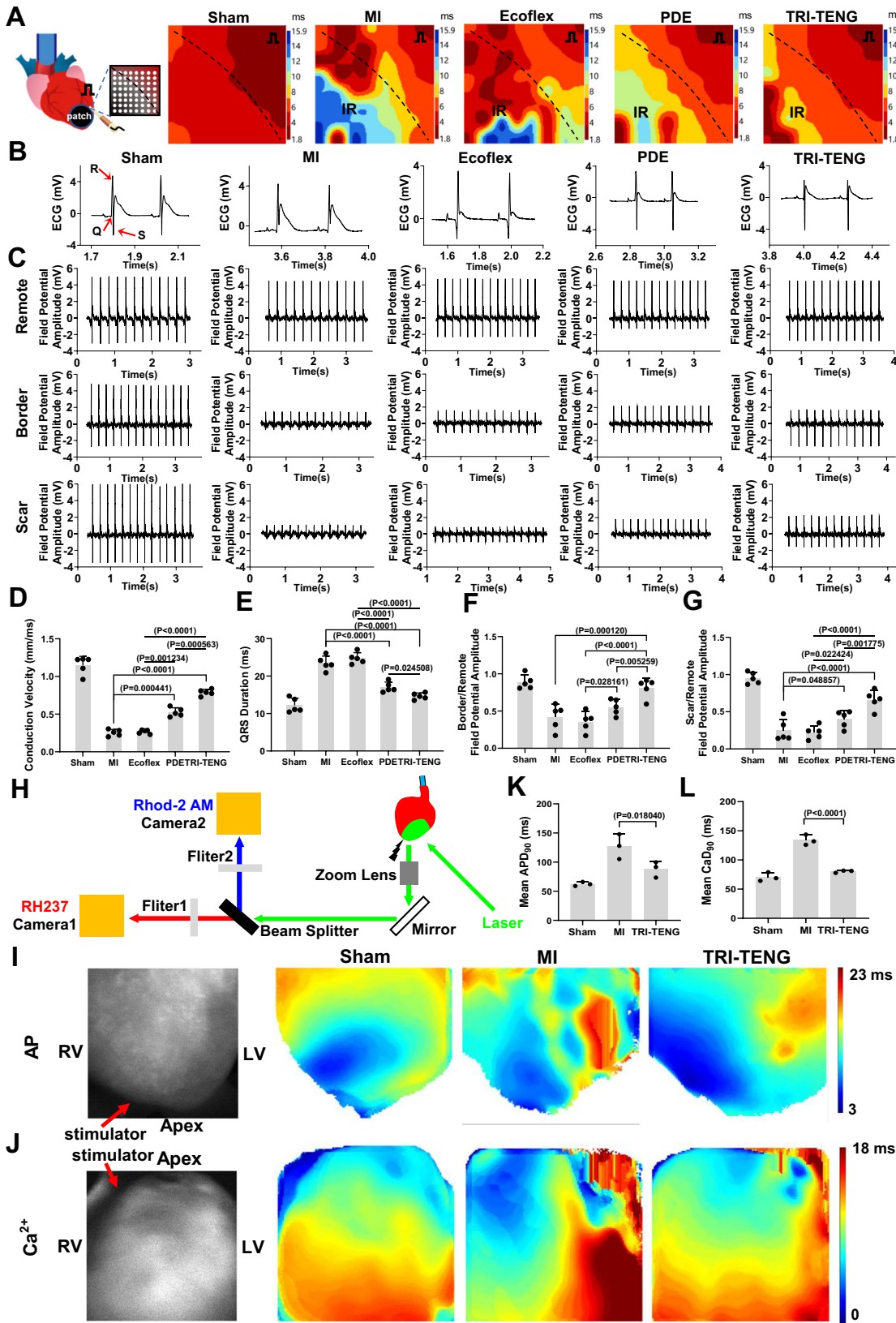

whereas Ecoflex transplantation had no effect in revascularization (Supplementary Fig. 18A, D, E). Notably, higher positive expression density of α-actinin/CX43 and vWF/α-SMA proteins in the infarct region were possessed in the TRI-TENG group compared to those in the other groups (Fig. 6D and Supplementary Fig. 18). Cardiac function assessments were carried out using echocardiography at 2 and 4 weeks

after the transplantation. The echocardiographic images revealed severe ventricular dilatation and stiff LV anterior wall activities in both the MI and the Ecoflex groups. Conversely, the PDE group exhibited weakened pumping action, and the TRI-TENG-implanted heart demonstrated apparent wall motion (Fig. 6E and Supplementary Movie 6). The quantitative echocardiographic data showed an increase

**Fig. 5 | Electrical mapping and optical mapping for the Langendorff-perfused hearts at week 4 after patches' transplantation. A** The schematic diagram of electrical mapping from Langendorff-perfused hearts in different groups at week 4 post-transplantation. The stimulating electrode was positioned inferior to the right atrial appendage, and the 64-channel electrode was placed at the border region. Representative epicardial activation maps in all groups were displayed. The dark lines demarcate the boundary between the non-infarcted and infarcted myocardium. IR means infarcted region. Red color indicates the earliest activation, while the blue color represents the latest activation. The numbers on the heatmap scale correspond to the time of activation in milliseconds. **B** Representative ECG traces of different groups. **C** The field potential amplitude of the remote region, border region and scar region in different groups at week 4 post-transplantation. **D** Conduction velocity of different groups calculated based on epicardial activation maps ($n = 5$ independent rats). **E** The QRS duration of different groups were

calculated based on ECG ($n = 5$ independent rats). **F, G** Statistics analysis of the border/remote field potential amplitude (**F**) and the scar/remote field potential amplitude (**G**) ratios in different groups ($n = 5$ independent rats). **H** Schematic of a setup for dual optical mapping of Rhod-2 AM-reported $Ca^{2+}$ transients and RH237-reported transmembrane voltage in Langendorff-perfused rat hearts. **I, J** Optical mapping images of action potential (AP) (**I**) and ventricular $Ca^{2+}$ transients initiation (**J**) from the Langendorff-perfused hearts in different groups at week 4 post-transplantation. RV right ventricle. LV left ventricle. **K, L** Comparison of AP durations at 90% repolarization ($APD_{90}$) and calcium transient durations at 90% recovery ($CaD_{90}$) averaged over the optical mapping field of view in different groups ($n = 3$ independent rats). The data were presented as mean ± SD. Statistical significance in (**D**) was calculated using two-sided one-way ANOVA with Dunnett's post hoc test, and statistical significance in (**E–G, K, L**) were calculated using two-sided one-way ANOVA with LSD post hoc test.

in Left ventricle internal diameter in diastole (LVIDd), Left ventricle internal diameter in systole (LVIDs) and a decrease in the fraction shorting (FS) and ejection fraction (EF), indicating the cardiac function deterioration in the MI and the Ecoflex groups (Fig. 6F). However, PDE and TRI-TENG transplantation improved cardiac function as evidenced by the decrease in LVIDs and the increase in ΔFS% and ΔEF% (Fig. 6G), indicating the improvement in cardiac pumping ability and a reduction in compensatory cardiac chamber enlargement. Compared to other groups, TRI-TENG group exhibited the highest ΔFS% and ΔEF%, resulting in the best cardiac function recovery. Collectively, we have demonstrated that the TCP, which incorporates electrical stimulation and conductivity in one CCP, is superior to the traditional CCP with a single property of conductivity in repairing damaged myocardium and improving cardiac function.

Given the structure and physiological function of porcine hearts are similar to those of human[5], we examined the TRI-TENG's treatment in minipig MI models after its transplantation for 4 weeks (Fig. 7A). The TRI-TENG array was developed to match the clinically relevant size of Bama minipig heart ($3 \, cm \times 3 \, cm$)[40]. At 4 weeks post-transplantation, the hearts were harvested for subsequent morphometric and histological analysis. The gross results indicated that the TRI-TENG array transplanted hearts exhibited a smaller infarct size and thicker anterior wall in comparison to those observed in MI hearts. (Fig. 7B and Supplementary Fig. 19). Masson's Trichrome staining revealed that the MI group formed a significant amount of fibrous tissue in the infarct region, whereas transplantation with TRI-TENG arrays reduced myocardial fibrosis and promoted neonatal myocardium formation (Fig. 7C). Consequently, the immunofluorescence staining results demonstrated that hearts transplanted with TRI-TENG featured more mature myocardial tissue, more abundant CX43 proteins, as well as higher densities of micro vessels ($vWF^+$ cells) and arterioles ($vWF^+/\alpha$-$SMA^+$ cells) in the infarct region (Fig. 7D and Supplementary Fig. 20). Furthermore, ex vivo electrical signal propagation in porcine fresh hearts of the sham, MI and TRI-TENG array groups, which were quickly removed and immediately immersed in Krebs–Henseleit solution (KH solution) to maintain activity, was evaluated by measuring ECG using a signal acquisition system (Fig. 7E). Under the same electrical signal stimulation, the local field potential amplitude of the MI group was attenuated obviously. However, the TRI-TENG array group produced about a fivefold increase in local field potential amplitude compared with the MI group (Fig. 7F, G). The enhanced electroactivity can augment the motility of the injured myocardium, thereby contributing to the amelioration of cardiac function. Therefore, echocardiography was conducted prior to MI induction and again 4 weeks post-operation in order to evaluate cardiac function variation in different groups. The stiff LV anterior wall activity appeared in the MI group, whereas the TRI-TENG array-implanted heart displayed obvious LV anterior wall activities (Fig. 7H and Supplementary Movie 7). The results revealed that the FS and EF values derived from short axis of left ventricular were reduced in the MI group at 4 weeks after the operation, while

increased significantly in the TRI-TENG array group compared with that in the MI group (Fig. 7I–L). We conducted a comparison of studies implementing intervention measures, including cardiac patch transplantation, gene engineering techniques, and delivery of bioactive factors, in porcine cardiac repair over the past five years, and discovered that the improvement effect for cardiac function by TRI-TENG array transplantation (FS improvement achieved about 14.7%) surpassed the efficacy of most other interventions (Supplementary Data 2).

To evaluate the potential toxicity and inflammatory response of TRI-TENG array transplantation in minipigs, blood samples in different groups were collected prior to MI induction, as well as at 2- and 4 weeks post-operation. In addition, vital organs including lungs, spleens, livers and kidneys from different groups were harvested at 4 weeks post-operation. The results of the routine blood tests indicated that there was no obvious difference among different groups (Supplementary Data 3). In addition, the results of blood biochemistry showed that the liver functions-related indicators and kidney functions-related indicators in the TRI-TENG array transplantation group have no significant difference compared with those in the sham group (Supplementary Data 4). Furthermore, the histological morphology of vital organs showed no significant alterations at 4 weeks after transplantation with TRI-TENG array, as evidenced by H&E staining (Supplementary Fig. 21). The in vivo inflammatory response to TRI-TENG array transplantation was detected by tracking the variation levels of IL-10, IL-1β, IL-6, and TNF-α cytokines at different points in time using ELISA. Researches have demonstrated that cytokines levels of IL-1β, IL-6, and TNF-α in the heart were persistently elevated after MI, and these elevated cytokines levels may exacerbate myocardial inflammation during the acute phase of MI[41,42]. Our findings indicate that the levels of IL-1β, IL-6, and TNF-α in the MI group were significantly elevated at 2 and 4 weeks post-operation compared to pre-operation. However, TRI-TENG array transplantation alleviated the upregulation of these cytokines at 2 and 4 weeks induced by MI. Besides, a remarkable increase of IL-10, which can relieve inflammation, was observed in TRI-TENG array transplanted hearts at 4 weeks, whereas there was no significant variation of IL-10 level in the MI group (Supplementary Fig. 22). Together, these results indicate that TRI-TENG array transplantation exhibits no apparent toxicity towards vital organs and can effectively suppress proinflammatory cytokines following MI.

## Whole-transcriptome RNA sequencing analysis of the change of gene level in different regions of infarct heart

TRI-TENG exhibits remarkable reparative effects on infarcted hearts of rats and pigs, primarily attributed to its capacity for enhancing the electroactivity of the infarcted myocardium. However, further investigation is required to elucidate the potential mechanisms at the genetic level. After transplantation for 4 weeks, rat's hearts of sham, MI, Ecoflex, PDE and TRI-TENG groups were harvested, and the tissues

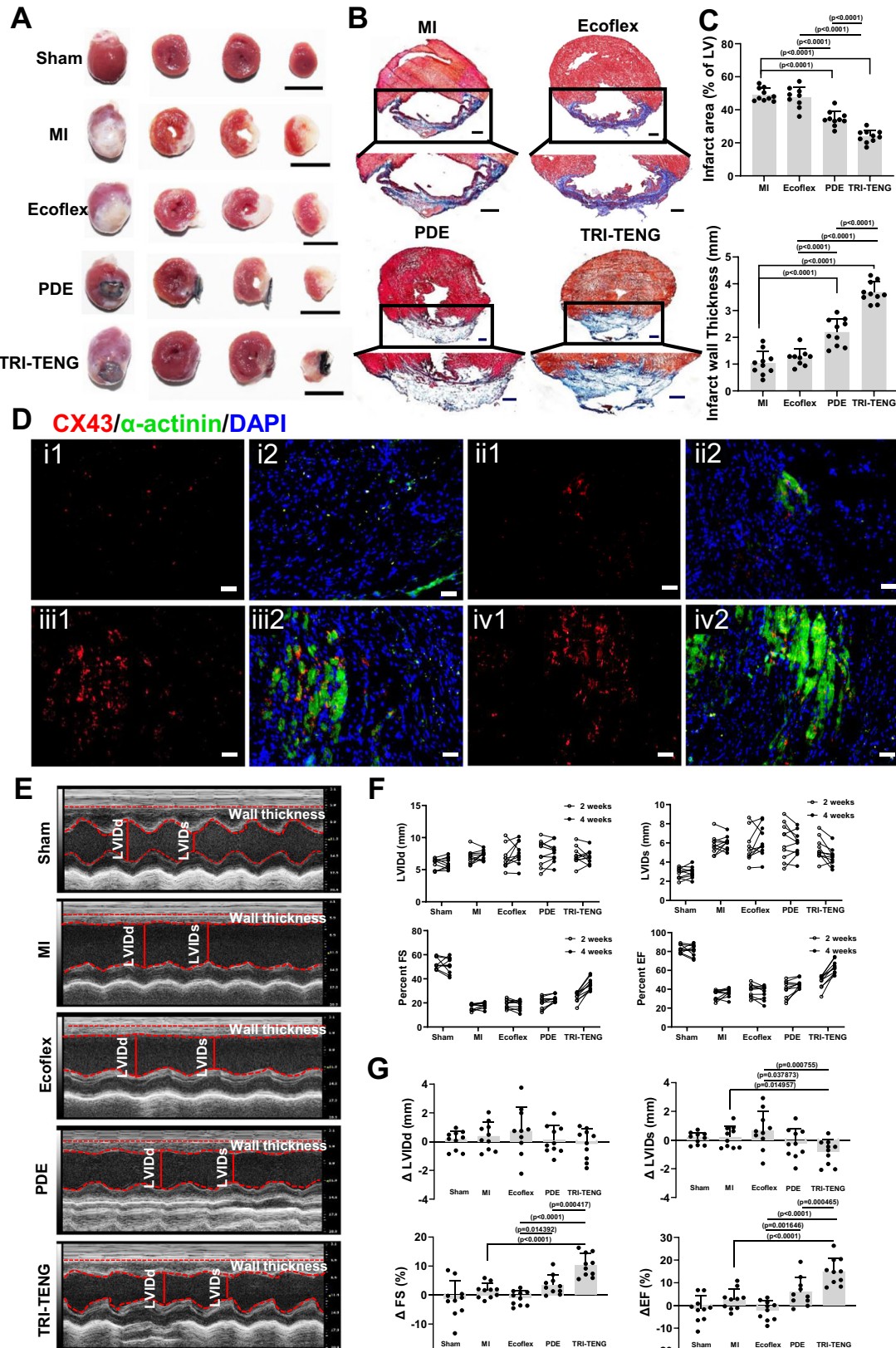

from infarct region (IR) and border region (BR) were dissected from the same heart for RNA sequencing (RNA-seq) to evaluate the changes in gene expression level. Principal component analysis (PCA) revealed that samples from the IR or BR of both Ecoflex and MI groups were tightly clustered, indicating minimal gene expression differences between Ecoflex and MI (Fig. 8A). In contrast, clear distinctions were

observed in the same regions between PDE and MI or TRI-TENG and MI, which pointed to their obvious gene expression differences. Besides, the pyramid diagrams and volcano plots depicting the number of differentially expressed genes (DEGs) in pairwise groups indicated that TRI-TENG group or PDE group exhibited a high degree of differences from MI group, while Ecoflex group and MI group shared

**Fig. 6 | The histological examination and assessment of cardiac function on rat hearts from different groups at week 4 after transplantation. A** Gross observations of the whole heart and three cross sections of the heart in different groups. Scale bars: 1 cm. **B** Masson's Trichrome staining for cardiac sections in different groups. blue: fibrosis tissue; red: myocardium. Scale bars: 1 mm. **C** Quantitative comparisons of the infarct area and infarct wall thickness among different groups. Each symbol denotes an independent rat (MI $n = 10$, Ecoflex $n = 9$, PDE $n = 10$, TRI-TENG $n = 10$). **D** Immunofluorescent staining of CX43 (red) and α-actinin (green) proteins in the infarct regions in different groups. Scale bars: 10 μm. i: MI group, ii:

Ecoflex group, iii: PDE group, iv: TRI-TENG group. **E** Echocardiograms of LV contraction in different groups at 4 weeks post-transplantation. **F** Changes of LVIDd, LVIDs, FS and EF in different groups determined by echocardiography at 2 and 4 weeks after transplantation ($n = 10$ independent rats). **G** Statistical results of the changes of the LVIDd, LVIDs, FS and EF in different groups measured by echocardiography ($n = 10$ independent rats). The data were presented as mean ± SD. Statistical significance was calculated using two-sided one-way ANOVA with LSD post hoc test.

more similar genetic perturbation trends (Fig. 8B, C, left in Supplementary Fig. 23A–F, and left in Supplementary Fig. 24A–F). Furthermore, the TCseq package was utilized to perform cluster analysis of perturbed genes, which revealed two distinct clusters (cluster 1 and cluster 2), and hierarchical clustering heatmaps of DEGs were constructed based on gene expression patterns of cluster 1 and cluster 2 (Fig. 8D–F). Hierarchical clustering heatmaps and trend diagrams revealed that genes in cluster 1 exhibited low expression in the sham, while these genes were upregulated in the MI compared with that in the sham. With different therapeutic measures, the expression of these genes showed a decreasing trend. Among them, TRI-TENG showed the largest downregulation in gene expression, converging towards the sham. The genes in cluster 2, on the other hand, exhibited high expression levels in the sham group and were downregulated in the MI group compared to those in the sham group. With different therapeutic methods, these genes displayed an increasing trend, and TRI-TENG showed the highest increase compared with that in the MI, which converged toward the sham.

To further investigate the mechanism by which TRI-TENG mediates the repair of infarcted myocardium, gene enrichment pathway analysis in both the IR and the BR in different groups was performed. Kyoto Encyclopedia of Genes and Genomes (KEGG) analysis of DEGs indicated that biological processes in the comparison of TRI-TENG vs MI were mainly enriched in energy metabolism-related pathways (citrate cycle, oxidative phosphorylation), $Ca^{2+}$ regulation-related pathways (calcium signaling pathways, MAPK signaling pathways), cardiac contraction-related pathways (cardiac muscle contraction, cAMP signaling pathway), cellular adhesion and spreading-related pathways (gap junction), vascular regulation-related pathways (vascular smooth muscle contraction), inflammatory regulation-related pathways (inflammatory mediator regulation of TRP channels), cell cycle regulation-related pathways (TGF-beta signaling pathway) (middle in Supplementary Fig. 23C and middle in Supplementary Fig. 24C). In the comparison of PDE vs MI, DEGs mainly appeared in $Ca^{2+}$ regulation-related pathways (PI3K-Akt signaling pathway), cellular adhesion and spreading-related pathways (ECM-receptor interaction, focal adhesion), energy metabolism-related pathways (PPAR signaling pathway, fatty acid degradation, citrate cycle, protein digestion and absorption), amino acid biosynthesis-related pathways (propanoate metabolism) and cardiac contraction-related pathways (cardiac muscle contraction) (middle in Supplementary Fig. 23B and middle in Supplementary Fig. 24B). In the comparison of TRI-TENG vs PDE, DEGs were found to be enriched in pathways associated with inflammatory regulation, energy metabolism, cell cycle and vascular regulation (middle in Supplementary Fig. 23F and middle in Supplementary Fig. 24F). Furthermore, gene ontology analysis revealed the enrichment of terms related to cardiac muscle contraction, wound healing, response to hypoxia, energy metabolism, cell cycle, angiogenesis, and ECM organization in the comparison of TRI-TENG vs MI (right in Supplementary Fig. 23C and right in Supplementary Fig. 24C). When comparing TRI-TENG group with PDE group, DEGs in the IR majorly occurred in wound healing, cell adhesion and organization, response to decreased oxygen levels, regulation of T cell activation and cell contraction (right in Supplementary Fig. 23F), and in the BR mainly enriched in cell cycle, ECM organization, regeneration and cell

proliferation (right in Supplementary Fig. 24F). In addition, the gene expression heatmap revealed a marked upregulation in the expression of genes associated with cardiac conduction (Dsc2, Hrc, Kcna5, Scn4b, Tnni3k), calcium handing (Cxcl11, Dhrs7c, Fhl2, Gbp1, Pdk2, Stc2), cardiac muscle contraction (Asb15, Atp2a2, Fgf13, Myh6, Scn5a, Tmem38a), angiogenesis (Angpt1, Igf2, Il1a, Rgcc, Vegfb) and wound healing (Celsr1, Gp1ba, Habp2, Klkb1, Ppara, Prkce) in the IR in the TRI-TENG and PDE groups compared with those in the MI and Ecoflex groups (Fig. 8G), and TRI-TENG group displayed the highest gene expression level among all groups. The results revealed significant alterations in genes associated with signaling pathways in the hearts transplanted with TRI-TENG or PDE, as compared to those in the MI hearts. These changes have a more effective regulatory impact on cardiac contraction, energy metabolism, ECM-receptor interaction, and inflammatory response, which reflects the molecular-level responses of PDE and TRI-TENG to the ischemic/hypoxic microenvironment and demonstrate their reparative effects on MI. Furthermore, the superior reparative effect of TRI-TENG compared to PDE is primarily attributed to its inherent self-powering characteristic of TRI-TENG. It can convert cardiac contractility into electrical signals to activate infarcted tissue. Specially, the TRI-TENG plays a pivotal role in enhancing cardiac conduction, calcium handling, energy metabolism, myocardial contractility, and angiogenesis, ultimately contributing to the repair and regeneration of infarcted myocardium.

## Methods
### Materials
GO was purchased from Hengqiu Tech., Inc. Hexane was purchased from Anachemia Canada, Inc. Polylactic acid (PLA) was purchased from XYZ printing. Ecoflex 00-50 was purchased from Smooth-On, Inc. 184 silicone elastomer kit was purchased from SYLGARD. DOPA was purchased from Sigma-Aldrich. PVDF was purchased from Tullagreen, Carrigtwohill, Co. Cork IRELAND. Tris-HCl and DMF were purchased from Sigma-Aldrich. Acetone was purchased from Thermo Scientific (USA). PVP was purchased from Shanghai Aladdin Bio-Chem Technology Co., LTD. Silver epoxy kit H20E was purchased from Epoxy Technology, Inc. The live/dead cell staining kit was purchased from Shanghai Bioscience Technology Co. Ltd. The primary antibodies of α-actinin, CX43 and vWF were purchased from Abcam (Britain). α-SMA antibodies was purchased from Bosterbio (USA). Alexa Fluor 568 donkey anti-rabbit IgG (H&L) and Alexa Fluor 488 donkey anti-mouse IgG (H&L) were acquired from Life Technologies (USA). Masson's Trichrome Stain Kit was purchased from Beijing Solarbio Science & Technology CO., LTD.

### Preparation of leaf vein template
Leaves were harvested from boldo plant (Peumus boldus). Cuticle and mesophyll cells of leaves were removed by immersing leaves in hexane for 1 day to acquire leaf vein templates.

### Preparation of molds
PLA round molds (diameter = 8 mm, height = 0.5 mm) were built for the preparation of rGO electrodes, PDA-rGO electrodes, and bottom package layer. PLA hollow cylinder molds (inner diameter = 6.5 mm, outer diameter = 8 mm, height = 5 mm) were built for the preparation

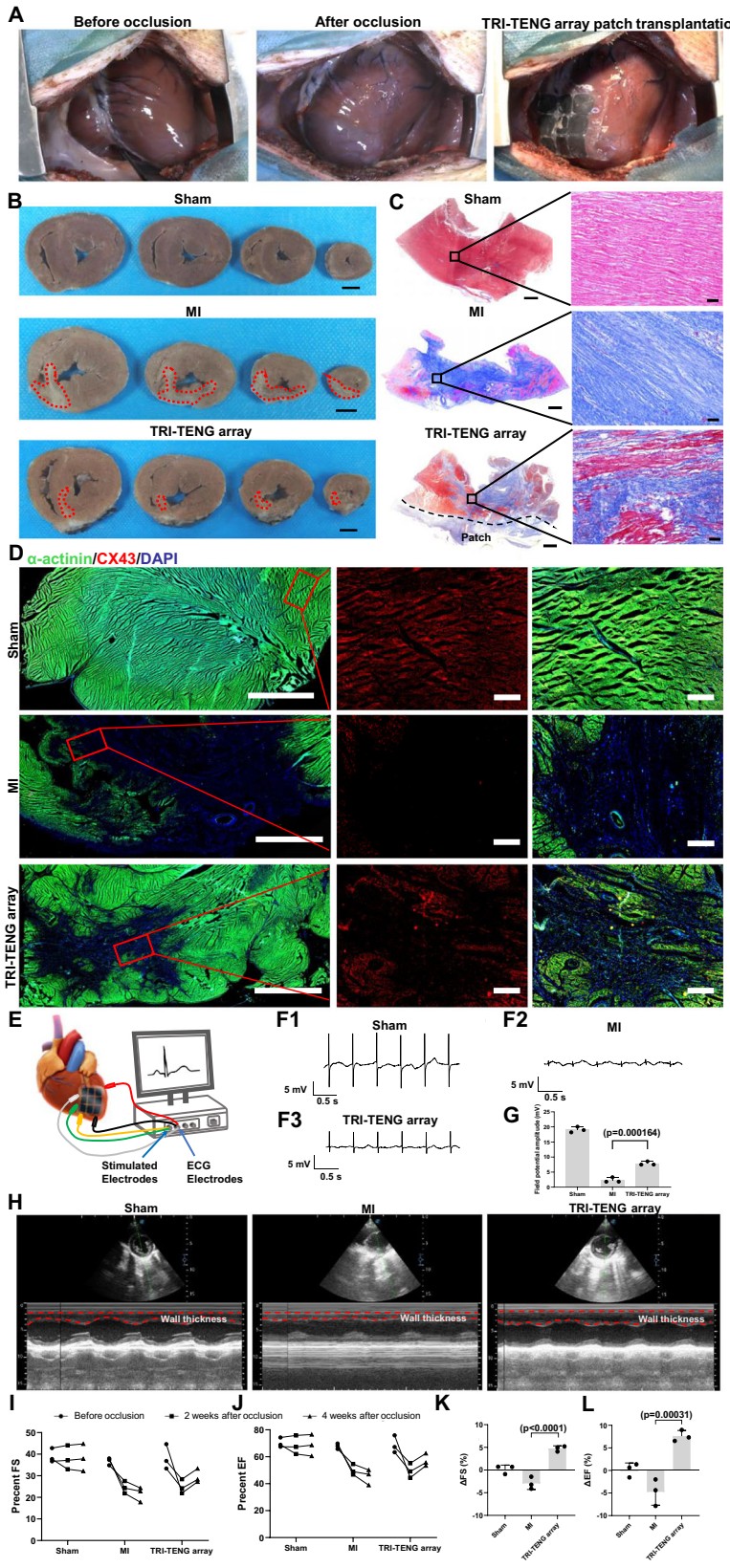

of Ecoflex spacers. All the molds were built using da Vinci Junior 1.0 3D printer (XYZ printing, Inc.). To prepare the leaf vein molds, two parts of 184 silicone elastomer kit were mixed at 10:1 ratio, and the mixture was spin-coated on a cover slid at 3500 rpm for 30 s. Then, the mixture was cured at 120 °C for 30 min to obtain a layer of cured elastomer.

Another layer of the mixture was spin-coated on top of the cured elastomer at 3500 rpm for 30 s after the cured elastomer is cooled down. After that, a piece of leaf vein template was placed in the uncured mixture. The uncured mixture was then cured at 120 °C for 30 min, and the piece of leaf vein template was torn off.

**Fig. 7 | The treatment effect of TRI-TENG array for the infarcted heart in porcine MI models. A** Representative macroscopic images of the minipig hearts pre- and post-occlusion, as well as following transplantation of the TRI-TENG array onto the heart. **B** Representative transverse sections of hearts from sham, MI and TRI-TENG array transplantation groups after 4 weeks. The heart was cut into four layers from apex to level of ligation. The scar region was highlighted in the sections. Scale bars, 2 cm. **C** Representative images showing myocardial fibrosis stained with Masson's Trichrome of cardiac sections in different groups. Scale bars, 2000 μm (left) and 50 μm (right). **D** The expressions of α-actinin (green) and CX43 (red) proteins in porcine cardiac sections from various groups. Scale bars, 5000 μm (left) and 500 μm (mid and right). **E** Schematic representation of the assessment of electrical

responses in infarcted myocardium in ex vivo porcine hearts using a signal acquisition system. **F** Representative ECG traces of the stimulation signal from ex vivo hearts in the sham (**F1**), MI (**F2**), TRI-TENG array (**F3**) groups. **G** The amplitude of the local field potential was subjected to statistical analysis across various groups ($n = 3$ independent minipigs). **H** Echocardiographic parasternal short-axis views of the minipigs' LV in different groups at the papillary-muscle level at 4 weeks after operation. **I, J** Cardiac function parameters, FS (**I**) and EF (**J**), in different groups before occlusion, as well as at 2 and 4 weeks post-operation. **K, L** Statistical analysis of the changes of FS (**K**), EF (**L**) in different groups at 4 weeks after transplantation ($n = 3$ independent minipigs). The data were presented as mean ± SD. Statistical significance was calculated using two-side one-way ANOVA with LSD post hoc test.

## Fabrication of rGO and PDA-rGO electrodes

GO powder was added in DI water and sonicated using Probe Sonicator (SK92-IIN) for 60 min to obtain an 8 mg/mL uniform GO solution. In total, 90 μL of the GO solution was drop casted on 8 mm PLA round molds and followed by evaporation at room temperature to get GO thin films. For 5 mm TRI-TENG, 18 μL of GO was drop casted on 5 mm mold. GO thin films were heated at 300 °C for 12 h to get rGO electrodes. DOPA was dissolved in Tris buffer with a pH of 8.5 to make 2 mg/mL DOPA solution. The rGO electrodes were immersed in the DOPA solution for 8 h for the formation of PDA coating on rGO electrodes. In all, 6 mm rGO and PDA-rGO electrodes were prepared in a similar manner using 6 mm PLA mold.

## Preparation of PVDF membrane with leaf vein structure

PVDF was dissolved in DMF/Acetone (2:1 ratio) under vigorous stirring for 30 min to obtain a homogeneous solution at the concentration of 10% (w/v). 150 L of the PVDF/DMF solution was cast on a leaf vein mold. A piece of PVDF membrane with a leaf vein structure was obtained after the solvent was vacuum-dried.

## Assembly of TRI-TENG and TRI-TENG array

PVP was dissolved in ethanol at 90 °C under vigorous stirring to make a 10% (w/v) ethanolic solution of PVP. The PVP/ethanol solution was spin-coated on a PLA round mold at 2000 rpm for 60 s. After the ethanol was evaporated, a PVP sacrificial layer was formed. Part A and part B of Ecoflex 00-50 were mixed at the ratio of 1:1. The mixture was spin-coated on top of the PVP sacrificial layer at the speed of 3500 rpm for 60 s. Curing of Ecoflex was performed at 60 °C for 30 min. Another Ecoflex mixture spin-coated on top of the previous Ecolex layer at the speed of 3500 rpm for 60 s. The rGO electrodes were placed on the uncured Ecoflex mixture followed by curing of the mixture. The PVDF membrane was cut into a circular shape with a diameter of 7 mm and was then placed on the rGO electrode. After 15 mg of the ecoflex mixture was drop cast onto the PLA hollow cylinder mold, the mold was placed on the PVDF layer with the uncured Ecoflex mixture facing the PVDF layer. The Ecoflex mixture was then cured at 60 °C for 30 min to make the Ecoflex spacer. The hollow cylinder mold was removed, and 5 mg of Ecoflex mixture was drop cast on the Ecoflex spacer. The PDA-rGO electrode was placed on top and the Ecofelx mixture was cured. The whole device was immersed in DI water and subjected to a water bath for 2 h for the detachment of TRI-TENG.

A 4 cm × 4 cm PLA mold was used as the initial spin-coating substrate for the TRI-TENG array. The 4 cm × 4 cm PLA molds were used for the preparation of rGO electrodes and PDA-rGO electrodes. A 4 cm × 4 cm grid PLA mold with a line width of 1.5 mm was used for the preparation of the Ecofelx spacer. A TRI-TENG configuration was used for the TRI-TENG array. The electrical connection was made between every two neighboring rGO electrodes as well as between every two neighboring PDA-rGO electrodes using air-dried PEDOT: PSS/GelMA hydrogel as solder. The GelMA was synthesized according to a previously reported technique[5]. In brief, 10 g of gelatin was dissolved in 100 mL PBS buffer at 50 °C under vigorous stirring to get a 10% (w/v)

gelatin aqueous solution, and then 2 mL of methacrylic anhydride (MAA) was added into the gelatin solution. The reaction proceeded at 50 °C for 2 h and the pH of the mixture was kept at 7.4 during the reaction using 10 M NaOH solution. The reaction product was dialyzed against DI water at 50 °C for 3 days and then lyophilized. In total, 100 mg of PEDOT: PSS, 100 mg of GelMA, and 2 mg of Irgacure® were dissolved in 2 mL DI water under vigorous stirring at 50 °C to obtain a homogeneous solution. A small amount (5–10 mg) of the precursor solution was placed on each connection point as solder. The precursor solution was then cured under UV for 30 min and air-dried. To make TRI-TENG with lead, rGO and PDA-rGO electrodes with a connection structure were made using PLA round mold with an extra connection portion and insulated wires were soldered on the connection structure using silver epoxy.

## Characterization of the TRI-TENG

The cyclic compressive mechanical input that drove the TRI-TENG and mechanical properties of the TRI-TENG was provided by MTS® universal testing machine. The electric signals were collected using a benchtop digital multimeter (Keithley DMM6500, Tektronix®). The TRI-TENGs with wire connected to benchtop multimeter was affixed to throat, finger, and ankle using scotch to evaluate the sensing capacity. In the in vitro characterization of the TENGs, the rGO electrode of TENG was connected to the digital multimeter. A 2 cm × 2 cm × 2 cm PDMS cube was affixed to the top compression platen of the universal testing machine. The contact and separation between the TRI-TENG and the PDMS cube were facilitated by cyclic compression performed by the testing machine.

FTIR was used to characterize the existence of specific functional groups. FTIR spectra were recorded using an infrared spectrophotometer (FTIR, Nicolet 6700, ThermoFisher Scientific Inc., USA). The sample spectra were recorded with a spectral range of 400–4000 cm$^{-1}$, a resolution of 4 cm$^{-1}$, and an average of 64 scans. Surface roughness of the rGO film and the PDA-rGO film was examined using an Atomic Force Microscope (AFM).

To prove that the sensing unit and stimulation do not interfere with each other, regardless of the complexity of in vivo voltage measurement, the folloeing setup was designed. A SA hydrogel that mimics the heart was placed on top of the first spacer layer of the TRI-TENG. Then the SA hydrogel was subject to cyclic compression to mimic the contact and separation of the PDA-rGO layer with the heart due to heartbeat, the rGO layer of the TRI-TENG was connected to a benchtop digital multimeter (Keithley DMM6514, Tektronix®) for the measurement of the voltage output of sensing unit. Additionally, the ECG electrodes were attached to the hydrogel to measure the electric potential built on the hydrogel by the stimulation unit.

## Experimental animals

Sprague-Dawley (SD) rats (newborns aged 1–3 days or adult weighing 250 ± 20 g) were purchased from the Animal Center of Southern Medical University, and Bama minipigs were purchased from Longgui Xingke Animal Breeding Farm, Baiyun District, Guangzhou City.

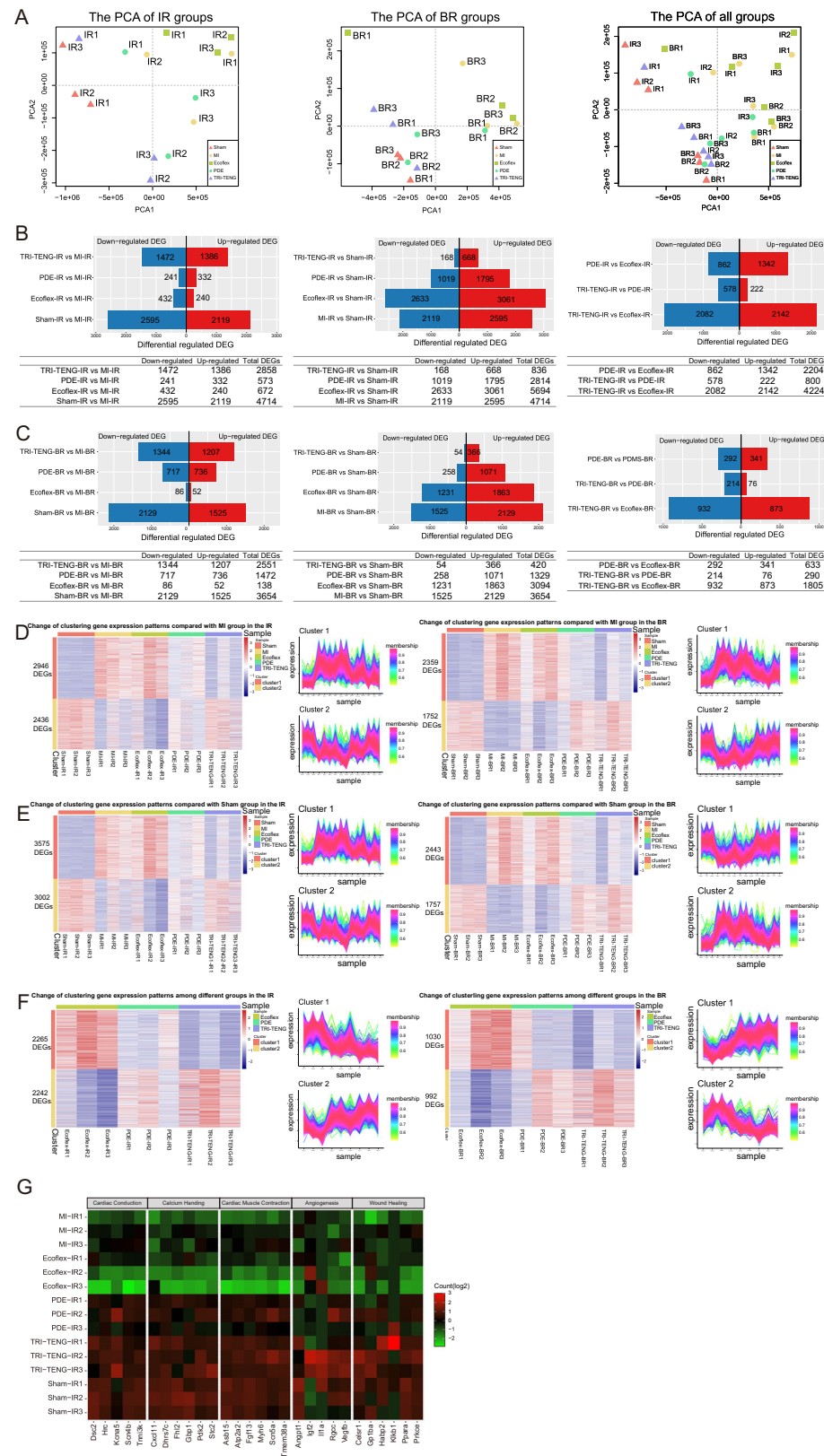

All animal experimental performed in accordance with the Regulations on the Administration of Laboratory Animals (China). The experiments in rats were approved by the Southern Medica University Animal Ethics Committee, and the experiments in minipigs were approved by the Experimental Animal Ethics Committee of Longguixingke Animal Farm.

## Sensing signal detection

After intramuscular administration of preanesthetic medication (atropine sulfate 0.05 ml/kg, xylazine hydrochloride 1 mg/kg), minipigs were anesthetized by continuous intravenous infusion of propofol (10 ml/h) and inhalation of isoflurane (1–3%). A thoracotomy was performed between the fourth and fifth ribs, then the fourth intercostal

**Fig. 8 | Gene expression pattern analysis for the infarct region (IR) and border region (BR) of rats in different groups at 4 weeks after transplantation by RNA-Seq. A** Principal component analysis (PCA) of the IR samples (left), the BR samples (middle) and all samples (right). **B, C** Differentially expressed genes (DEGs) in each pairwise comparison in the IR (**B**) and the BR (**C**). Upper: pyramid diagrams of the number of DEGs. Red represents the number of upregulated genes, blue represents the number of downregulated genes. Lower: the tables of the number of downregulated genes, upregulated genes and total DEGs. **D** Clustering trend analysis based on the union of differential genes comparing the groups (sham, Ecoflex, PDE and TRI-TENG) vs the MI group in the IR (left) and the BR (right). **E** Clustering trend analysis based on the union of differential genes comparing the groups (MI, Ecoflex, PDE and TRI-TENG) vs the sham group in the IR and the BR. **F** Clustering trend analysis based on the union of differential genes comparing TRI-TENG vs Ecoflex, TRI-TENG vs PDE and PDE vs Ecoflex in the IR and the BR. Left: Hierarchical clustering heatmap, right: gene expression trend diagram in the (**D–F**). **G** Gene expression heatmap of the specified function-related genes in different groups. Green, low expression. Red, high expression.

space was widened by a rib spreader, and the pericardium was opened and TRI-TENG patch with wire was placed between the ventricle and the pericardium. Voltage signal was recorded using the multimeter (DMM7510, Keithley) attached to the wire and ECG was recorded by signal acquisition system (BL-420F, Chengdu Techman Software Co. LTD).

Rats were injected with heparin (3125 U/kg) for 10 min to prevent blood and humanely sacrificed using the pentobarbital sodium. The heart was excised quickly and rinsed in Krebs–Henseleit (KH) buffer solution containing 128 mM NaCl, 4.7 mM KCl, 20 mM NaHCO$_3$, 1.05 mM MgCl$_2$, 1.19 mM NaH$_2$PO$_4$, 1.3 mM CaCl$_2$ and 11.1 mM D-glucose. Then the heart was perfused retrograde in Langendorff apparatus with KH solution bubbled with 95% O$_2$ and 5% CO$_2$, and stabilized for 15 min at $37 \pm 0.5\,°C$. The flow rate and the perfusion pressure were maintained at 8–10 ml/min and 60–70 mmHg, respectively. A TRI-TENG patch (0.8 cm in diameter) with wire was placed to the left ventricle, and the multimeter was connected to the wire to record the voltage signal of the TRI-TENG driven by the heart under normal and ischemic conditions, respectively. Then an electrical stimulus with 7 Hz stimulation was performed under the left atrial appendage and the voltage signal was recorded. Pulses was performed by an electronic stimulator. ECG electrodes were positioned on the right atrium and the left ventricle, respectively, to continuously record the ECG.

In order to evaluate the effect of reduced cardiac contractile motion in the ischemic region on the TRI-TENG' function, a small size TRI-TENG (0.6 cm in diameter) was prepared. ECG signals obtained from Langendorff-perfused normal and ischemic rat hearts before and after the TRI-TENG (without wire) transplantation were recorded. Besides, the TRI-TENG with wire was placed to the left ventricle, and the multimeter was connected to the wire to record the voltage signal of the TRI-TENG driven by the normal and ischemic hearts, respectively.

Male SD rats were anesthetized, performed ventilation and then carried out thoracotomy for wireless sensing experiment. The TRI-TENG with a wire was connected to an external wireless Bluetooth device (Pokit meter). To be specific, when the TRI-TENG was placed on the heart, its wire was connected to the electrodes of the Bluetooth device, which was positioned outside the chest. Then the TRI-TENG was placed on the rat heart to collect the V$_{OC}$ under normal and ischemic conditions. After receiving the wireless signal, the waveform of the V$_{OC}$ was displayed on a mobile phone. ECG signal was collected by signal acquisition system simultaneously. In addition, in order to evaluate the effect of rat's breathing movement on sensing function of the TRI-TENG, the output voltage of the TRI-TENG and the corresponding ECG signals under both open-chest and closed-chest conditions were recorded. Male SD rats were anesthetized and carried out thoracotomy. Then, the TRI-TENG was directly transplanted on the exposed heart to harvest biomechanical energy from the heart's beating under the open-chest and the closed-chest conditions, respectively. ECG signal was collected by the signal acquisition system simultaneously.

### Neonatal rat CMs culture
The hearts of 1–3-day-old neonatal SD rats were quickly collected. After the hearts were carefully dissected, the heart tissues were washed three times with PBS solution to remove the blood clots and dissociated in trypsin overnight at 4 °C, followed by digesting into single-cell suspension with 0.1% collagenase type II (Sigma). The isolated cells were pre-plated for 2 h to remove cardiac fibroblasts and the nonadherent cells were seeded on the different scaffolds ($5 \times 10^6$ cells/cm$^2$). CMs were cultured in high-glucose Dulbecco's modified Eagle's medium (DMEM, GIBCO) supplemented with 15% fetal bovine serum (FBS, GIBCO), 100 U/ml penicillin, and 100 μg/ml streptomycin. All cells were maintained in an incubator at 37 °C under 5% CO$_2$ and the culture mediums were exchanged every 2 days.

### Biocompatibility evaluation
The biocompatibility of TRI-TENG was tested using live/dead staining. After CMs were cultured on Ecoflex, PDE and TRI-TENG for 1 day, 3 days, 7 days, respectively, the cells were washed three times with PBS and stained with the live/dead working solution at 37 °C for 5 min in the dark. The images of dyed samples were acquired with a laser scanning confocal microscope (LSM 880, Zeiss). Cell viability was defined as the ratio of living cells to total cells. The number of cells was quantified using Image J software (v2.0.0) at two independent sites of each sample, and each sample was prepared three times.

### Scanning electron microscopy (SEM)
The microstructure of different scaffolds and the micromorphology of CMs seeded on scaffolds were observed by SEM. After being cultured for 7 days, the CMs-loaded scaffolds were rinsed with PBS, fixed with glutaraldehyde at 4 °C overnight, dehydrated with ethanol and dried by the critical point method. Then they were photographed under a SEM (ULTRA55, Zeiss). All samples were treated by spray-gold.

### Immunofluorescence staining for the cultured cells
For immunofluorescence staining, CMs were co-cultured with different matrixes for 3 days and 7 days, respectively. The samples were fixed with 4% paraformaldehyde (PFA) at 4 °C overnight and washed three times with PBS. Then samples were permeabilized with 0.2% Triton X-100 solution for 15 min at room temperature and blocked with 2% bovine serum albumin (BSA)/PBS for 30 min at room temperature. The samples were incubated with the primary antibodies Mouse anti-α-actinin (1:250) plus Rabbit anti-CX43 (1:200) diluted in 2% BSA/PBS at 4 °C overnight. The primary antibodies were removed and the samples were incubated in secondary antibodies for 2 h at room temperature. The secondary antibodies applied were as follows: Alexa Fluor 488 donkey anti-Mouse IgG (H&L) (1:500) plus Alexa Fluor 568 donkey anti-Rabbit IgG (H&L) (1:500). The F-actin (Yeasen Biotechnology (Shanghai) Co., Ltd.) status was analyzed by phalloidin staining with fluorescein-isothiocyanate-conjugated phalloidin (1:500) for 1 h at room temperature. All samples were stained with DAPI (Santa Cruz) and imaged under a fluorescence microscope (BX53, Olympus).

### Implantation of different cardiac patches into rat MI model
Male SD rats were served as MI model and divided into the sham group, MI group, the Ecoflex group, the PDE group and the TRI-TENG group. In brief, all rats were anesthetized, performed ventilation and then carried out thoracotomy. Rats in the sham group underwent thoracotomy, and other rats underwent ligation operations of the left

anterior descending artery (LAD) after thoracotomy. Electrocardiograph monitoring showed a remarkable elevation of ST segment, indicating a successful MI model of rat. Fifteen minutes after ligation, Ecoflex, PDE and TRI-TENG were implanted respectively on the surface of the infarcted myocardium, and the edges of patches were sutured on the epicardium of the border of the infarcted myocardium with 7-0 polypropylene sutures. To further validate the safety and impact of the TRI-TENG on normal heart's function and structure, the TRI-TENG patch was implanted on the normal heart.

## Cardiac stimulus threshold and contraction force of ventricular tissue

Rats were given heparin to prevent clotting, humanely killed by cervical dislocation and the hearts were quickly removed. Then the health heart was stabilized by perfusion for 15 min to discharge the residual blood until the cardiac activity resumed normal rhythm, and ECG was monitored. Pulses were performed by an electronic stimulator under the left aurcle, and the pacing voltage was started at 1 V and increased in 0.5 V increments up to achieve ECG capture. The ECG was recorded as the stimulus threshold of the normal heart. After applying Ecoflex, PDE and TRI-TENG to the left ventricle, respectively, the ECG signals were recorded as the stimulus threshold of the transplanted heart. For further examining the effects of different patches on stimulus threshold after cardiac ischemia, cardiac ischemia was produced by ligation of the LAD, which made ST-segment elevation in ECG. Then Ecoflex, PDE and TRI-TENG were applied. ECG signals of ischemic and transplanted hearts were recorded.

The heart was quickly carried out, then intact ventricular tissue was isolated in KH buffer, secured between two vascular clamps, and contracted by 1 Hz pulse of the ventricular apex. First, the contractility of ventricular tissue was recorded using the mechanical sensing module of signal acquisition system. Then, Ecoflex, PDE and TRI-TENG were attached to the ventricular tissue in sequence, and their contraction force were recorded. Four weeks after patches (Ecoflex, PDE and TRI-TENG) transplantation in rats, the ventricular tissues in different groups were collected and tested for contraction force by the same method.

## Electrical mapping and optical mapping

Rats treated with heparin were sacrificed. Lungs were clamped when hearts were exposed. After that, the hearts were lifted and cut quickly along the back of the lungs. Hearts were then Langendoff-perfused with KH buffer solution with the flow rate of 10 ml/min at $37 \pm 0.5$ °C. After the residual blood was discharged, hearts were perfused and monitored for stability for 15 min before experimental procedures commenced. The 64 separated electrodes ($8 \times 8$ grids, 0.55 mm spacing) were placed at the border region between the healthy and infarct myocardium. A 2 mV 5 Hz pulses of electrical stimulus was applied to the epicardium right under the left atrial appendage. ECG electrodes were positioned on the right atrium and left ventricle respectively, pacing from the right atrium and continuously recording the ECG.

Hearts were rapidly collected and perfused in KH solution containing 10 μM blebbistatin used to stop contractions and avoid movement artefacts. Dye loading was aided by pre-perfusion with pluronic F-127 (20 % w/v in DMSO). The calcium dye Rhod-2-AM (1 mg/ml) and the voltage-sensitive dye Rh237 (1 mg/ml) were added to the perfusion solution in sequence to measure membrane potential and $Ca^{2+}$. The heart was illuminated by 530 nm fluorescence light using LEDs (MGL-III-532-100mw) and imaged by the 50-mm camera to record voltage signals. Emission lights were collected at >700 nm for Rh237 and $590 \pm 15$ nm for Rhod-2-AM. The emitted fluorescence signals were recorded using two CMOS cameras (01-KINETIX-M-C, TeledynePhotometrics), $350 \times 350$ pixel spatial resolution for a sampling rate of 100 Hz.

## Echocardiography evaluation of rat cardiac function

The left heart function of all animal groups was evaluated by IE33 echocardiography (Vevo2100, Visual Sonics). At two weeks and four weeks after patch transplantation, the rats were anesthetized, and transthoracic echocardiography was performed. M-mode echocardiography was recorded with a 40-MHz transducer, and short-axis views were obtained to measure the cardiac parameters. The cardiac functional parameters, including length of Left ventricle internal diameter in diastole (LVIDd), length of Left ventricle internal diameter in systole (LVIDs), Fraction Shorting (FS), and Ejection Fraction (EF) were measured.

## Morphology, histology, and immunofluorescence assay for cardiac sections

At week 4 after implantation, the rats were sacrificed, and the hearts were collected. The harvested hearts were sliced into three sections and immediately fixed in 4% paraformaldehyde overnight at 4 °C. Images of the heart cross-sectional gross morphology were captured. Then the slices were dehydrated in 15% and 30% sucrose, respectively, and frozen in OCT at −20 °C. The cryosections of 6 μm in thickness were prepared and placed onto slides for histological analysis. The infarct area and wall thicknesses were defined by the Masson's Trichrome staining. The infarct size was determined by the ratio of the inner perimeter of the scar region and the entire inner perimeter of the ventriculus sinister wall. The thicknesses of the scar region were measured three times and averaged. The immunostaining for cryosections used the same method as above. The primary antibodies of Rabbit anti-vWF antibody (1:200) plus Mouse anti-α-SMA (1:100), Mouse anti-α-actinin (1:250) plus Rabbit anti-CX43 (1:200) were used. The secondary antibodies applied were as follows: Alexa Fluor 488 donkey anti-Mouse IgG (H&L) (1:500) plus Alexa Fluor 568 donkey anti-Rabbit IgG (H&L) (1:500). All samples were imaged in a fluorescence microscope and the images were analyzed by Image J software.

## Minipigs model of MI and TRI-TENG array implantation

Minipigs were kept in the constant temperature of 22–25 °C for 1 week to adapt the environment and were randomly divided into the Sham group, MI group and TRI-TENG group. The pigs were anesthetized by propofol (10 ml/h) and isoflurane (1–3%), and then subjected to thoracotomy. The fourth intercostal space was widened to visualize the LAD and the LAD was ligated with a 5-0 polypropylene suture (Prolene, Ethicon) for 10 min, reperfused twice and permanently tied. MI models were confirmed by ST-segment elevation on electrocardiogram and cyanosis of the myocardial surface during the operation. In the Sham group, the same thoracotomy was performed without the LAD ligation. Then the chest of MI group was sutured, and the infarcted myocardium area of TRI-TENG group was covered with TRI-TENG array. Electrocardiogram, body temperature, blood pressure and arterial oxygen saturation were monitored throughout the operation. Body weight and physiological status was monitored daily.

After 4 weeks, ex vivo electrical signal propagation in the infarcted area was evaluated by measuring ECG using a signal acquisition system. After euthanizing the pig, the hearts were removed and immersed in KH solution. One end of the infarcted area was connected to stimulating electrodes, and used ECG electrodes to detect electrical signals by the two-lead method. The output was set for 2 Hz pulse.

## Echocardiography of minipigs

Echocardiography of minipigs was obtained using a portable echocardiograph before occlusion, at 2 and 4 weeks after operation. The anesthetized minipigs were placed in the left lateral decubitus position and covered with warmed ultrasound gel in the chest. The echocardiography was acquired to measure LVIDs, LVIDd, FS and EF from M-mode tracings at the mid-papillary level.

## Histology and immunofluorescence of tissue sections of minipigs

The pigs were euthanized at 4 weeks postoperatively. A small part of organs (lung, liver, spleen and kidney) was removed, washed with PBS and fixed with 10% formalin. The whole hearts were quickly excised and balloons with 20 ml of formalin were filled into the hearts to prevent the chamber collapse of the left ventricle. Then, the hearts were cut into four 1 cm-thick slices from the apex to the atrium and fixed with 10% formalin. After fixation for 48 h, the hearts were completely dehydrated in 20% sucrose solution for 24 h and 30% sucrose solution for 24 h in a 4-degree refrigerator. For Masson's trichrome staining and immunostaining analysis, the procedure was the same as described above. The tissue sections of the lung, liver, spleen and kidney were embedded in paraffin and stained with hematoxylin and eosin (H&E) for pathological examination.

## Collection and testing of blood samples

To assess the potential toxicity and inflammation in the porcine infarcted hearts resulting from TRI-TENG array transplantation, blood samples were collected before MI induction, 2 weeks after operation and 4 weeks after operation. Blood routine was tested using anticoagulant, and blood biochemistry and ELISA were tested using serum. Blood samples were tested by Guangzhou Huayin Medical Laboratory Center Co., Ltd.

IL-1β, IL-6, TNF-α, and IL-10 levels of blood samples were measured by ELISA. Liver and kidney functions were examined by measuring porcine blood levels of aspartate aminotransferase (AST), alanine aminotransferase (ALT), aspartate aminotransferase/alanine aminotransferase (AST/ALT), γ-glutamyl transpeptidase (γ-GT), lactate dehydrogenase (LDH), alkaline phosphatase (ALP), total protein (TP), albumin (ALB), globulin (GLB), albumin/globulin (A/G), total bilirubin (TBIL), direct bilirubin (DBIL), indirect bilirubin (IBIL), creatine (CREA), UREA, and glucose (GLU).

## RNA sequencing of heart tissues and bioinformatics analysis

The rats were euthanized at 4 weeks after MI, and heart tissues were rapidly collected from the infarct region and the border region. RNA extraction and quality examination were performed by Annoroad Gene Technology Co. Ltd. (Beijing, China). RNA degradation and contamination were monitored on 1% agarose gels. RNA purity was checked using the NanoPhotometer® spectrophotometer (IMPLEN, CA, USA). The integrity of RNA samples was assessed using the RNA Nano 6000 Assay Kit of the Bioanalyzer 2100 system (Agilent Technologies, CA, USA). Each 2 μg RNA sample was used as input material for the RNA sample preparations. Sequencing libraries were generated using NEBNext® UltraTM RNA Library Prep Kit for Illumina® (NEB, USA) following the manufacturer's recommendations. The clustering of the index-coded samples was performed on a cBot Cluster Generation System using TruSeq PE Cluster Kit v3-cBot-HS (Illumia) according to the manufacturer's instructions. After cluster generation, the library preparations were sequenced on an Illumina Novaseq platform.

When analyzing data, reference genome and gene model annotation files were downloaded from genome website directly. Gene Ontology (GO) enrichment analysis of differentially expressed genes was implemented by the clusterProfiler R package, the statistical enrichment of differential expression genes in KEGG pathways was tested by clusterProfiler R package.

## Statistical analysis

The data were expressed as mean ± SD at least three independent experiments. Comparisons between two groups were performed using independent samples $t$ test and in multiple groups were performed using one-way ANOVA with LSD post hoc test to determine the significance of differences. $P$ value less than 0.05 was considered statistically significant. Statistical analyses were compared using SPSS 23.0 software. The average value and s.d. for all graphs were plotted using GraphPad Prism v.8.0.

## Reporting summary

Further information on research design is available in the Nature Portfolio Reporting Summary linked to this article.

## Data availability

All data to support the results in this study are included in the paper and supplementary information. The raw and analyzed datasets are available from the corresponding authors upon request. Source data are provided with this paper.

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

## Acknowledgements

This work was supported by the National Natural Science Foundation of China (Grant No. 32071355 and 31922043 to L.W.), the National Key R&D Program of China (2022YFC2402801 to L.W.), Science and Technology Planning Project of Guangdong Province (2022A1515011888 to L.W.).

## Author contributions

L.W. and M.X. conceived the research, supervised the project and provided research direction. R.Q., X.Z. and C.S. synthesized and characterized the materials. M.X., L.W., R.Q., X.Z., C.S. and K.X. performed the in vitro experiments. R.Q., X.Z., C.S., H.N., Y.L., X.X., Q.L. and X.S. performed the in vivo experiments. M.X., L.W., R.Q., X.Z., C.S., K.X., H.N., Y.L., X.X., K.M., Q.Y., Q.L. and X.S. verified data integrity and performed the statistical analyses. M.X., L.W., R.Q., X.Z., C.S., K.X., H.N., Y.L., X.X., K.M., Q.Y., Q.L. and X.S. interpreted the data and co-wrote the manuscript. All the authors reviewed the manuscript.

## Competing interests

The authors declare no competing interests.
