## [Peer Review File · Nature Communications]

REVIEWER COMMENTS

Reviewer #1 (Remarks to the Author):

This is an interesting work, very well-developed and very well-organized. I enjoy reading it, and I would like to say congrats authors for this amazing work. I have a few questions and suggestions before publication.

The authors would like to highlight the reason for choosing this type of layered structure with rGO and PDMS in this work. As far as i know, it would be better to use a metal-organic framework (MOF) and MXene in this regard, but maybe the authors want to explore it in the future, so it's not a negative comment. But. I will encourage authors to describe their reason of choosing this type of layered material.

Also, it would be better to provide more in-depth discussion in each part. Right now, the technological development in this work is amazing, but there is not much in-depth discussion and comparison with the literature.

Reviewer #2 (Remarks to the Author):

1. The PVDF layer is mold-cast to create a leaf vein structure. Such a structure may have a randomized pattern if it mimics naturally occurring leaves. Why is such a pattern chosen over others, such as a regular grid pattern of pyramidal or pin structures? It would help to show the performance improvement of leaf pattern electrodes over other artificial patterns.
2. It is mentioned in the manuscript that the TENG device is connected to a Bluetooth device for wireless data transfer. However, would this Bluetooth device also be implanted into the subject? If so, how would It be encapsulated and supported within the subject to ensure other bodily functions are not impeded?
3. What is the total output power of the E-cardiac patch, and what is the power input for electrical stimulation? How does the self-powered energy output match the stimulation power requirement?
4. What is the motivation for using PDA-rGO and PVDF as counterparts?
5. How does the PDMS spacer affect the flexibility and effectiveness of the E-cardiac patch?
6. PDMS and EcoFlex are not the same material. Please provide consistent information on the device throughout the text.

7. As the authors state, triboelectric sensors are affected by subtle mechanical stimuli. In this sense, considering living subjects, how do the mechanical stimuli derived from the individual's motion affect the charge transferred to the heart? Furthermore, how does this motion-induced change in output affect the synchronous rhythmic heartbeat? Please provide experimental evidence.
8. The references used by the authors regarding the triboelectric series (25,26) do not include epicardium or PDA-rGO. Please remove the claim about these material's tendency to lose or gain electrons. Instead, provide experimental evidence based on the current flow during contact separation.
9. Why does the current produced by the TRI-TENG excite the injured heart but do not affect a healthy heart?
10. The authors claim that the TRI-TENG has 80% cell viability after three days. Please compare this with other reported values in the literature for other biocompatible implantable materials.
11. Always use the complete form of a word before introducing its abbreviation in the text. Many abbreviations were used before being introduced, like LV.
12. It is claimed that the double spacer structure is unique. However, this multilayer design has been published elsewhere (ACS Nano 2013, 7, 4, 3713–3719). Fig. 1 should explain clearly what the difference is.
13. Why are the voltage outputs positive and negative (Fig. 2)? Discuss it analytically.

Reviewer #3 (Remarks to the Author):

This manuscript describes the development of a novel strategy for creating an electrically conductive and self-sensing cardiac patch to facilitate the regeneration of injured myocardium after myocardial infarction (MI). The researchers employed a concept based on a trinity triboelectric nanogenerator (TRI-TENG), using reduced graphene oxide (rGO) and biomimetic patterned polydopamine (PDA) to generate suspendable and sandwiched sheets (placed between PDMS spaces) for generating an electric field based on the triboelectric characteristics of these layers. The manuscript presents a wealth of data, both in vitro and in vivo, covering material characterization and the testing of the proposed system's capabilities in small and large animal models to investigate the hypothesis that the proposed triboelectric nanogenerator can stimulate injured myocardium during the contractility cycles and facilitate tissue regeneration. The data demonstrates the effectiveness of the proposed system in myocardial regeneration in both small and large animal models. Additionally, mechanistic studies were conducted using RNA-seq analyses to elucidate the role of specific upregulated pathways and gene signatures that may contribute to enhanced regenerative outcomes.

Despite significant amount of data, there are still questions yet to be explored and addressed in this work:

- It's important to note that the concept of a triboelectric nanogenerator has been previously utilized by both the authors and other investigators in the field of cardiac regeneration. While the concept itself is not entirely novel, this manuscript introduces a new patch system design built upon this established concept along with extensive amount of data.

- The authors have not investigated the degradation of the PDMS and its potential influence on the functionality of the proposed system in vivo. Most of the testing has been conducted over a four-week period, and it remains unclear whether the proposed system, once the PDMS degrades, will remain functional for a longer duration, such as 12 weeks. This is an important consideration, as many previous translational studies on myocardial infarction (MI) have assessed the efficiency of engineered patches for more extended periods, typically up to three months.

- The data presented in Figure 2 need to be tested and generated for long periods of cycles. Perhaps an automated stretching/flexing set up would help the authors to test the capabilities, durability and reproducibility of the proposed system for long duration of cycles.

- In their gene expression analyses, the investigators have shown upregulated pathways using bulk RNA sequencing. However, a question that remains unanswered is why the authors did not employ single-cell RNA sequencing to examine how the phenotypic signatures of cardiomyocytes, cardiac fibroblast cells, and potentially endothelial cells influence the tissue regeneration process as well as vascularization when exposed to the proposed patch.

- The sensing capabilities of the proposed patch were demonstrated in only one of the studies, using the Langendorff setup. Therefore, it raises the question of why the authors did not evaluate the self-sensing capabilities over the long term, particularly using the mini-pig studies.

Reviewer #4 (Remarks to the Author):

This manuscript describes a E-cardiac patch of miniature self-powered biomimetic trinity triboelectric nanogenerator (TRI-TENG) for sensing and repairing infarcted myocardium. The authors demonstrated that this device improved the electroactivity of the infarcted heart, including the excitation-contraction coupling, Ca²⁺ transient and action potential (AP) propagation. Furthermore, the authors successfully revealed the patch could also wirelessly monitor electrocardiosignal to the mobile device for diagnosis. RNA sequencing analysis from rat hearts revealed that E-cardiac patch mainly regulated cardiac muscle contraction-, energy metabolism-, vascular regulation-related mRNA expression in vivo. While the in vivo characterization and presentation of the work is good it is not clear what the impact of the E-cardiac patch is. It is critical for the implantable device to have degradable properties for short-term service in vivo. Thus, novelty is limited. In addition, more explains are also needed to strengthen authors' statements.

1. The nanogenerator can convert mechanical energy into electrical signal/electricity. Notwithstanding, it is unrealistic for a single device to achieve both functions (sensing and electrical stimulation repair) at the same time. Are you conducting separate experiments to verify the sensing and treatment functions of the E-cardiac patch? Or does a single patch have both sensing and treatment functions?
2. Due to myocardial cell necrosis at the site of myocardial infarction, cardiac systolic and diastolic function will decrease in this area. Therefore, can the E-cardiac patch still convert cardiac biomechanical energy into electrical energy well in the area of myocardial infarction? The authors need to add a follow-up experiment to demonstrate the actual electrical performance of the device in the myocardial infarction region compared to normal heart tissue.
3. The manuscript lacks any information regarding the reproducibility of the findings. How many animals did you use? How many measurements of voltage, current, and charge were made? Why choose two animal models, swine and rat? There is no necessary connection between the electrical performance in large animals and the actual effects in small animals.
4. One of the key elements of this work is the novel myocardial patch. The authors need to present a physical picture of the E-cardiac patch. In addition, the authors should clarify the actual dimensions of the devices used in the different experiments
5. The edges of patches were sutured on the epicardium of the border of the infarcted myocardium with 7-0 polypropylene sutures. Does the suturing process have a negative impact on the myocardial infarction area? In addition, myocardial infarction repair is a relatively short-term treatment process. After the E-cardiac patch repairs MI, will the device be removed through a second surgery?
6. Fig. 1A- the third column shows the healing mechanism that an electrical potential built between the PDA-rGO electrode and the myocardium upon the contraction and relaxation of the myocardium. What does each rectangular unit represent? The authors need to rephrase it clearly.
7. Fig. 2A. The PDA coating on the PDA-rGO electrode further increased the open-circuit voltage, short-circuit current, and transferred charges to 21.98 mV, 2.23 nA, and 0.22 nC, respectively. The authors should further explain the specific reasons.

8. Why are the structures involved in device performance characterization in vitro not the same as in vivo? In vivo testing of nanogenerators will be seriously affected by environmental interference. The electrical output of the device in the body is in the order of mV and nA. How exactly did the authors measure it?

9. By watching the video in the supplementary material, we found that the device is still relatively large and hard. Are the same devices used in long-term in vivo experiments?

10. Are you running wires from the electrodes to test the electrical output of the E-cardiac patch? Does the device still have a wire structure during the process of repairing myocardial infarction? The authors need to further clarify how the E-cardiac patch works in vivo.

11. Minor comments

i. >Figure 2B: Please modify the expression of the ordinate - "Current (nA) (nA)"

ii.>The picture clarity of the manuscript needs to be improved.

Reviewer #1 (Remarks to the Author):

This is an interesting work, very well-developed and very well-organized. I enjoy reading it, and I would like to say congrats authors for this amazing work. I have a few questions and suggestions before publication.

1. The authors would like to highlight the reason for choosing this type of layered structure with rGO and PDMS in this work. As far as I know, it would be better to use a metal-organic framework (MOF) and MXene in this regard, but maybe the authors want to explore it in the future, so it's not a negative comment. But. I will encourage authors to describe their reason of choosing this type of layered material.

Answer: Thank you for highlighting the superiority of MXene over rGO. The material for the electrode layer must be conductive, biocompatible, biodegradable, and able to be fabricated into a membrane with the desired structure through mold casting. The material for the package should be biocompatible, water-impermeable, and amenable to mold casting.

The choice of rGO and polybutylene adipate terephthalate (PBAT) Ecoflex™ stems from the fact that these two materials meet the aforementioned requirements perfectly. The rGO was chosen in this study because of its high conductivity and ease of fabrication into a membrane with any desired structure through mold casting/vacuum-assisted filtration. In addition, its biocompatibility can be significantly improved by PDA coating. In our previous study (Adv. Funct. Mater. 2023, 33 (32), 2300866), PDA-coated rGO has been proven to facilitate cardiomyocytes' (CMs') adhesion, growth, and viability. It also decreased the infarcted area and improved the vascularization of rat models with myocardial infarction (MI). PBAT Ecoflex™ was employed because it is also subject to mold casting for the preparation of desired geometry. Moreover, it is biocompatible and approved by the Food and Drug Administration (FDA) for contact with food. Furthermore, the monomers in PBAT are bonded via ester linkages, making PBAT highly biodegradable. Polydimethylsiloxane (PDMS) has been extensively used as a material for molds in the fabrication of microneedles, microfluidic systems, and microarrays. It is transparent, flexible, easy to fabricate, and capable of capturing extremely small structural features. Thus, we

utilized PDMS to bestow the surface of the PVDF triboelectric layer with a leaf vein structure.

The MXene is a superior alternative for rGO. The intrinsic hydrophilicity makes it more biocompatible and biodegradable than rGO (Adv. Healthc. Mater. 2019, 8, 1801137. Sciences 2002, 99, 10287-10292. Chem. Eng. J. 2020, 400, 126009). The flexibility and stretchability of pure rGO film is limited. Though the inherent conductivity of rGO is high, it cannot improve the conductivity of the composite as effectively as the MXene when it serves as filler in the composite (J. Mater. Chem. 2012, 22, 8512-8517. PNAS 2014, 111, 16676-16681). The free-standing PVA/MXene thin film can be simply fabricated through vacuum-assisted filtration, achieving a conductivity of 2.2×10^4 S/m. It exhibits high flexibility and strong hydro-stability (Nano Energy 2021, 84, 105921), making it a perfect fit for the in vivo environment of the epicardium. In addition, the PVA/MXene thin film can be structured through laser cutting and photolithography. We will use it in our future study.

2. Also, it would be better to provide more in-depth discussion in each part. Right now, the technological development in this work is amazing, but there is not much in-depth discussion and comparison with the literature.

Answer: Thank you for your suggestion. We have made a table (Supplementary Table 1) that summarizes the literature that develops TENGs for the healthcare of the cardiac system. We have added Supplementary Table 3 in the revised supporting file to make a comprehensive comparison of the biocompatibility of our TRI-TENG cardiac patch with other implantable self-powered materials, as reported in existing literature. Besides, we have provided more discussions on our TENG technology in the introduction and discussion.

Reviewer #2 (Remarks to the Author):

1. The PVDF layer is mold-cast to create a leaf vein structure. Such a structure may have a randomized pattern if it mimics naturally occurring leaves. Why is such a pattern chosen over others, such as a regular grid pattern of pyramidal or pin structures? It would help to show the performance improvement of leaf pattern electrodes over other artificial patterns.

Answer: Thank you very much for your comments. Various nanostructures such as nanopillars, nanowires, nanobowls, nanopyramids, and nanoparticles have been employed in surface modification strategies to increase the surface roughness of the triboelectric layer and improve the performance of TENG (Adv. Mater. Technol-us 2021, 6, 2000916). The nanopillar and nanopyramid patterns gained favor among scholars due to their output, high uniformity, and excellent reproducibility (Nano Energy 2015, 17, 63-71). Templating is the most facile and cost effective way to fabricate a substantial amount of triboelectric layers with nanostructures. However, this method heavily relies on patterned silicon (Si) molds, the fabrication of which involves high-cost and intricate processes such as photolithography and etching. Nano/microstructures in nature including lotus leaves, rose petals, and cicada wings, have been adopted to surfaces to improve the performance of TENG (Small 2014, 10, 3887-3894). The molds derived from nature offer a more cost-effective and accessible alternative patterned Si molds. To minimize randomization, a consistent leaf vein was used exclusively throughout this study to fabricate patterned triboelectric layers. Crumpling is another affordable and easily obtainable alternative. However, the resulting crumpled pattern exhibits greater levels of randomization (Nano Energy 2018, 46, 73-80. Nano Energy 2019, 58, 304-311). Overall, accessibility and cost are the main considerations guiding our choice of the leaf vein structure. Leaf pattern shows advantages of cost-effective and easy to obtain; also, leaf pattern increases the surface roughness that makes the triboelectric device effectively contact the heart.

2. It is mentioned in the manuscript that the TENG device is connected to a Bluetooth device for wireless data transfer. However, would this Bluetooth device also be implanted into the subject? If so, how would It be encapsulated and supported within the subject to ensure other bodily functions are not impeded?

Answer: Thanks for your comment. In the current stage of our research, we have primarily focused on validating the sensing capabilities of the TRI-TENG and creating a functional prototype. The prototype we designed incorporates TRI-TENG with a wired connection to an external Bluetooth device. Specifically, once the TRI-TENG is positioned on the heart, its wire is attached to the electrodes of the Bluetooth module, which is located outside the chest. This configuration enables the device to capture the output voltage from the TRI-TENG and wirelessly relay the signals to a mobile phone for immediate monitoring.

We acknowledge the importance of creating a more integrated and patient-friendly system for long-term applications. As such, we are considering making the Bluetooth device more compact and implantable in future iterations, facilitating sustained, real-time monitoring.

To provide a comprehensive understanding of our methodology and address your concerns, we have added a detailed description in the part of “Sensing Signal Detection” of the method in the revised manuscript. We believe these modifications and clarifications will significantly improve the manuscript and appreciate your constructive feedback.

3. What is the total output power of the E-cardiac patch, and what is the power input for electrical stimulation? How does the self-powered energy output match the stimulation power requirement?

Answer: Thanks for your comment. The application of the programmed electrical stimulation at the twice diastolic threshold to survivors of MI has been studied (J Am Coll Cardiol. 1991, 18, 780-788), and the timing of the programmed electrical stimulation has been studied (J Am Coll Cardiol. 1986, 8, 1279-1288). However, electrical stimulation that is high enough to pace the heart led to adverse effects (Cardiovasc Res. 2001, 49, 127-134). It is suggested that long-term low-amplitude electric pulses applied to myocardial infarct scar can prevent myocardial infarct thinning. To prevent the adverse effects and any effects on a healthy heart, electrical stimulation at a non-excitatory voltage (~ 0.25 mV) was adopted in this study (Fig 3I

in the revised manuscripts). The *in vivo* energy output of our E-cardiac patch is 2.317 nW/m² (Fig. 1 below).

Fig. 1 Electrical performance of TRI-TENG *in vivo*. n=10.

4. What is the motivation for using PDA-rGO and PVDF as counterparts?

Answer: Thanks for your comment. The material for the electrode layer must be conductive, biocompatible, biodegradable, and able to be fabricated into a membrane with the desired structure through mold casting. The PDA-rGO was chosen in this study because of its high conductivity and ease of fabrication into a membrane with any desired structure through mold casting/vacuum-assisted filtration. In addition, its biocompatibility can be significantly improved by PDA coating. In our previous study (Adv. Funct. Mater. 2023, 33 (32), 2300866), PDA-coated rGO has been proven to facilitate cardiomyocytes' (CMs') adhesion, growth, and viability. It also decreased the infarcted area and improved the vascularization of rat models with MI. Moreover, the PDA coating can increase the roughness of rGO in a cost-effective manner.

We initially selected PDA-rGO as the electrode layer, the counterpart for PDA-rGO was decided afterwards. The desired characteristic of the counterpart for PDA-rGO is to have a position as negative in the triboelectric series as possible, as GO is positioned relatively positively in the triboelectric series (Adv Mater 2018, 30, e1801210). In addition, the counterpart should be easily able to fabricate into a soft membrane with unique surface structures through mold casting. Thus, we chose PVDF as the counterpart for PDA-rGO.

5. How does the PDMS spacer affect the flexibility and effectiveness of the E-cardiac patch?

Answer: Thank you for your comments. We apologize for any confusion caused by the

incorrect labeling of the Ecoflex 00-50 as PDMS. The thickness of the E-cardiac patch is mainly dictated by the height of the Ecoflex 00-50 spacer. As the Ecoflex spacer height increases, the E-cardiac patch becomes, leading to reduced flexibility of the patch. The reduced flexibility compromises the conformity and healing efficacy of E-cardiac patch. The height of the Ecoflex 00-50 spacer defines the gap distance between the triboelectric layer in the E-cardiac patch. Theoretically, the performance of contact-separation mode TENG increases with an increase in the gap distance between triboelectric layer when the gap distance is smaller than 1 mm. However, it decreases when as the gap distance exceeds 1 mm (Nat. Commun. 2015, 6, 8376). The height of the Ecoflex 00-50 spacer was set at 1mm, as it results in the best TENG performance and is not too thick to affect the flexibility of the E-cardiac patch. Thus, the effectiveness of the E-cardiac patch can be optimized.

6. PDMS and EcoFlex are not the same material. Please provide consistent information on the device throughout the text.

Answer: Thank you for your suggestion. In this research, we used polybutylene adipate terephthalate (Ecoflex 00-50) as a component of the device, and PDMS was used as mold for the preparation of patterned PVDF. We have corrected all PDMS into Ecoflex throughout the text in describing the device components.

7. As the authors state, triboelectric sensors are affected by subtle mechanical stimuli. In this sense, considering living subjects, how do the mechanical stimuli derived from the individual's motion affect the charge transferred to the heart? Furthermore, how does this motion-induced change in output affect the synchronous rhythmic heartbeat? Please provide experimental evidence.

Answer: Thanks for your comment. When the TRI-TENG works as a sensor in chest, the breathing motion of the chest may make an effect on its function. To investigate this, we conducted experiments to measure the output voltage of the TRI-TENG and the corresponding ECG signals under both open-chest and closed-chest conditions.

Fig. 2A illustrates our setup: Under the open-chest condition, the TRI-TENG was directly transplanted on the exposed heart, with the lungs retracted, to harvest biomechanical energy from the heart's beating. Under the closed-chest condition, we

considered additional factors such as lung and chest motion that could potentially affect the TRI-TENG's sensing capabilities. Our findings revealed no significant difference in the open-circuit voltage generated by the TRI-TENG between the two conditions, both in terms of amplitude and frequency. This suggests that the external chest movements, including those from breathing, do not substantially alter the sensing capability of the TRI-TENG.

Furthermore, to assess the impact on the synchronous rhythmic heartbeat, we recorded the ECG signals in both scenarios. As depicted in Fig. 2B, D, the heartbeat frequency obtained from the ECG under open-chest conditions were consistent with those under closed-chest conditions, indicating that the chest motion does not significantly affect the heart's rhythmic activity.

Fig. 2 Electrical signal outputs of TRI-TENG and ECG signals recorded from rat hearts under both open-chest and closed-chest conditions. (A) Representative macroscopic images of TRI-TENG's electrical output assessment in rats under different conditions. (B) Open-circuit voltage (V_{oc}) values recorded from a rat heart transplanted with TRI-TENG under various conditions. The number of peaks in V_{oc} and in ECG

marked by red arrows under open-chest conditions corresponds to those under closed-chest conditions. (C, D) Statistical analyses of the open-circuit voltage difference (ΔV_{OC}) (C) and heart rate (D) from the rat hearts (n=3 each group. The data were presented as mean \pm SD).

8. The references used by the authors regarding the triboelectric series (25,26) do not include epicardium or PDA-rGO. Please remove the claim about these material's tendency to lose or gain electrons. Instead, provide experimental evidence based on the current flow during contact separation.

Answer: Thank you for your suggestion. We have removed the claim about these material's tendency to lose or gain electrons. Instead, we provided experimental evidence based on the sign of voltage output during contact and separation by using the benchtop digital multimeter (Keithley DMM6514, Tektronix[®]) (as shown below). To assess the PDA-rGO's tendency to lose or gain electrons, contact and separation between the PDA-rGO and the copper electrode-attached PVDF were made, and the voltage output was measured by connecting the copper electrode to the positive port of the multimeter. As shown in Fig. 3A, separating the PVDF and the PDA-rGO results in a negative voltage, indicating that the positive charges are driven from the ground to the copper electrode. Thus, the triboelectric charges in the PVDF are negative. The PDA-rGO tends to lose electrons compared to PVDF. Similarly, to assess the tendency of the myocardium to lose or gain electrons, separation and contact between the copper electrode-attached PDA-rGO and PVDF were performed with the copper electrode connected to the positive port of the multimeter (Fig. 3B). When the myocardium is separated from the PDA-rGO, a negative voltage is generated, indicating there is a flow of positive charges from the ground to the copper electrode. As the flow of positive charges is induced to balance the electrical field built between the PDA-rGO and the myocardium, the generated triboelectric charges on PDA-rGO must be negative. Thus, the PDA-rGO tends to gain electrons compared to the myocardium. Overall, the order of tendency to gain electrons is PVDF > PDA-rGO > myocardium.

Fig. 3 Voltage measured from a copper electrode (A) attached to the PVDF contacting with and separating from PDA-rGO and (B) attached to the PDA-rGO contacting with and separating from the myocardium.

9. Why does the current produced by the TRI-TENG excite the injured heart but do not affect a healthy heart?

Answer: Thanks for your suggestion. Under sinus rhythm, the mean V_{oc} generated by TRI-TENG from the rat healthy heart was about 0.25 mV, which falls within the range of the non-excitatory stimulation applied on rat heart (0.2~0.6 mV), and is insufficient to induce ectopic pacing (Peptides 2015, 65, 46-52). Consequently, the healthy heart

remains largely unaffected due to its higher threshold for electrical excitation.

In contrast, for the injured hearts, the damaged myocardium, lose electrical connectivity with the surrounding healthy tissue, could respond to external electrical stimulation, which is beneficial to the repair of MI (J Thorac Cardiovasc Surg, 2018, 156(2), 568-575). Our results demonstrated that electrical stimulation produced from the TRI-TENG patch at the injured site, combined with conductivity, can improve electrical conduction in the scar region and strengthen ventricular contractility. This effect is advantageous for MI repair and can ultimately improve the cardiac function of the infarcted heart.

To further validate the safety and impact of TRI-TENG on normal heart function and structure, we conducted additional animal experiments. We transplanted TRI-TENG patch onto the normal heart, and assessed cardiac function using echocardiography at 4 weeks after transplantation, followed by histological examination using Masson's trichrome staining at 4 weeks (Fig. 4). The results from Masson's trichrome staining for cardiac sections exhibited that the myocardium in the TRI-TENG transplantation group have no significant difference compared with those in the normal group. Similarly, the echocardiographic assessments showed no significant differences in contraction wave, pumping function, or quantitative parameters such as LVIDd, LVIDs, EF, and FS between the normal and TRI-TENG transplantation groups. These findings indicated that TRI-TENG transplantation did not adversely affect the structure or function of a healthy heart.

Fig. 4 Histological examination and assessment of cardiac function in rat hearts from the normal group and the TRI-TENG transplanted group at week 4 post-operation. (A) Masson's Trichrome staining for cardiac sections in two groups. Red: myocardium. Scale bars: 1 mm. (B) Echocardiograms of left ventricular (LV) contraction in two groups. (C) Values of Left ventricle internal diameter in diastole (LVIDd), Left ventricle internal diameter in systole (LVIDs), Ejection Fraction (EF) and Fraction Shorting (FS) determined by echocardiography in two groups at week 4 post-operation (n=5 each group. The data were presented as mean ± SD).

10. The authors claim that the TRI-TENG has 80% cell viability after three days. Please compare this with other reported values in the literature for other biocompatible implantable materials.

Answer: We are grateful for the opportunity to enhance our manuscript with additional comparative data. We have added Supplementary Table 3 in the revised supporting file. This table provides a comprehensive comparison of the biocompatibility of our TRI-TENG cardiac patch with other notable implantable self-powered materials, as reported in existing literature. These materials are commonly employed in a range of medical applications, including wound healing, tumor therapy, bone defect repair, and promoting the maturation of cardiomyocytes *in vitro*. It's noteworthy to mention that

the majority of these studies reveal a high level of biocompatibility, typically exceeding 70%. Given that the CM survival rate in the TRI-TENG group exceeded 80% on both day 3 and day 7 of culture, we concluded that the TRI-TENG patch meets the biocompatibility benchmarks of current self-powered implantable materials.

Supplementary Table 3. The cell viability of different implantable self-powered materials.

Implantable self-powered materials	Cell viability	Application	Ref.
TRI-TENG	Over 80%	Myocardial infarction repair	Our work
Injectable triboelectric nanogenerator (I-TENG)	Over 90%	Wound healing	¹
Composed of a self-powered triboelectric nanogenerator (TENG) and an implantable nitric oxide (NO)	Over 90%	Intracranial neuroglioma therapy	²
Bioresorbable natural-materials-based TENG (BN-TENG)	Over 95%	Improving dysfunctional cardiomyocyte contraction in vitro	³
Composed of TENG and interdigitated electrode	Over 80%	Promoting the maturation of cardiomyocytes in vitro	⁴
Piezoelectric acrylate epoxidized soybean oil (AESO) scaffolds doped with piezoelectric Ag-TMSPM-pBT (ATP) nanoparticles (AESO-ATP scaffolds)	Over 80%	Bone regeneration	⁵
Composite scaffold consisting of BaTiO ₃ coated on porous Ti6Al4V	Over 80%	Repairing bone defects	⁶

CaCO ₃ -mineralized piezoelectric biodegradable scaffolds based on two polymers: poly[(R)3-hydroxybutyrate] (PHB) and poly[3-hydroxybutyrate-co-3-hydroxyvalerate] (PHBV)	Over 70%	Stimulating the growth of bone tissue	7
Injectable T-BTO-nanoparticles-embedded thermosensitive hydrogel	Over 80%	Tumor eradication	8
Dynamically evolving nanocomposites	Over 80%	Wound healing	9

- [1] Xiao X, Meng X, Kim D, et al. Ultrasound-Driven Injectable and Fully Biodegradable Triboelectric Nanogenerators[J]. *Small Methods*, 2023: 2201350.
- [2] Yao S, Zheng M, Wang Z, et al. Self-Powered, Implantable, and Wirelessly Controlled NO Generation System for Intracranial Neuroglioma Therapy[J]. *Advanced Materials*, 2022, 34(50): 2205881.
- [3] Jiang, W. et al. Fully bioabsorbable natural-materials-based triboelectric nanogenerators. *Adv. Mater.* 30, 1801895 (2018).
- [4] Zhao, L. et al. Promoting maturation and contractile function of neonatal rat cardiomyocytes by self-powered implantable triboelectric nanogenerator. *Nano Energy* 103, 107798 (2022).
- [5] Li G, Li Z, Min Y, et al. 3D-Printed Piezoelectric Scaffolds with Shape Memory Polymer for Bone Regeneration[J]. *Small*, 2023: 2302927.
- [6] Liu W, Li X, Jiao Y, et al. Biological effects of a three-dimensionally printed Ti6Al4V scaffold coated with piezoelectric BaTiO₃ nanoparticles on bone formation[J]. *ACS applied materials & interfaces*, 2020, 12(46): 51885-51903.
- [7] Chernozem R V, Surmeneva M A, Shkarina S N, et al. Piezoelectric 3-D fibrous poly (3-hydroxybutyrate)-based scaffolds ultrasound-mineralized with calcium carbonate for bone tissue engineering: inorganic phase formation, osteoblast cell adhesion, and proliferation[J]. *ACS applied materials & interfaces*, 2019, 11(21): 19522-19533.

[8] Zhu P, Chen Y, Shi J. Piezocatalytic tumor therapy by ultrasound-triggered and BaTiO₃-mediated piezoelectricity[J]. *Advanced Materials*, 2020, 32(29): 2001976.

[9] Zhu Z, Gou X, Liu L, et al. Dynamically evolving piezoelectric nanocomposites for antibacterial and repair-promoting applications in infected wound healing[J]. *Acta Biomaterialia*, 2023, 157: 566-577.

11. Always use the complete form of a word before introducing its abbreviation in the text. Many abbreviations were used before being introduced, like LV.

Answer: We appreciate your thoughtful reminder. We understand the importance of clear communication and have, therefore, thoroughly reviewed our document to ensure all abbreviations are properly introduced. The complete forms of abbreviated words have been incorporated before their respective abbreviations and are highlighted in blue in the revised manuscript for easy identification. We believe these changes improve the readability of the text.

12. It is claimed that the double spacer structure is unique. However, this multilayer design has been published elsewhere (*ACS Nano* 2013, 7, 4, 3713–3719). Fig. 1 should explain clearly what the difference is.

Answer: Thank you for your comments. Several multilayer TENGs have been published (*ACS nano* 2013 7, 3713-3719. *ACS nano* 2018 13, 698-705. *Nat. Commun.* 2016 7, 12985). However, the multilayer TENG in this study distinguishes itself from through both a novel design and means of application.

The multilayer TENG in *ACS Nano* 2013, 7, 4, 3713–3719 is fabricated through depositing metal electrodes (A) and dielectric layer-attached metal electrodes (B) on both sides of a zigzag shaped Kapton film. As Fig. 5A of *ACS Nano* 2013 paper shows, the metal electrodes and dielectric layer-attached metal electrodes are arranged in an ABAB alternative structure. The two metal electrodes on the two sides of the Kapton film have different potential, resulting in the formation of a parasitic capacitor with the Kapton film serving as the dielectric layer. In this study, however, the PVDF and myocardium exhibit symmetry about the PDA-rGO electrode (Fig. 5C). The top and bottom surfaces of the PDA-rGO have the same potential, eliminating the presence of parasitic capacitance.

The TENG design in this study bears a closer resemblance to the symmetrical TENG design discussed in ACS Nano 2019, 13, 698-705, as shown in the Fig. 5B. In the symmetrical TENG design, metal electrodes on both sides of the Kapton film share the same potential, thereby avoiding the parasitic capacitance effect. The TENG design in this study can be viewed as fusion of the metal electrodes and eliminating the Kapton film in the symmetrical TENG design. Moreover, both the alternate and symmetrical TENG contain more than two repeating units, with the overarching goal scaling up electrical output while maintaining a constant TENG area.

A similar double spacer design was adopted in a three-layer TENG (Nat. Commun. 2016, 7, 12985), as illustrated in Fig. 5D, E. In the three-layer TENG design, a middle electrode layer with the bottom surface grounded was inserted between the top electrode-attached dielectric layer and the bottom electrode layer to enhance the electrical output. The contact of each layer occurs sequentially, while the separation happens simultaneously. The middle layer takes advantage of electrophoresis resulting from electrostatic induction, ensuring spatial separation of negative and positive charges within the middle electrode layer. Notably, there is no triboelectrification between the bottom surface of the middle electrode layer and the bottom electrode layer.

In this study, spatial separation of negative and positive charges within the middle PDA-rGO electrode also occurs due to electrostatic induction. However, all the contacts occur simultaneously with both the top and bottom surface of the middle PDA-rGO electrode undergoing triboelectrification, and the bottom surface of the middle electrode layer is not grounded. In our design, the myocardium is integrated into and functions as one component in TENG, enabling the bottom surface of the PDA-rGO electrode to directly establish an electric field on the myocardium. Myocardium is inferior in triboelectric charges generation. When it contacts with the bottom surface, very few triboelectric charges would be generated, resulting in a low potential between the bottom surface and the myocardium. Thanks to the electrostatic induction, the simultaneous triboelectrification between the top surface of the PDA-rGO electrode and another dielectric layer with a greater ability to generate triboelectric charges than myocardium makes the potential built between the bottom surface and inferior

myocardium equals that built between the top surface and superior dielectric layer.

We have made a new scheme depicting the healing mechanism in Fig. 1C in the revised manuscripts which clearly explains the difference (Fig. 5C below). In addition, the rGO electrode that is attached to the top dielectric layer was connected with a wireless measuring system in this study to achieve wireless monitoring of the myocardium. Overall, the purpose of utilizing multilayer design in TENG is to enhance the electrical output. It's worth noting that the principle of enhancing electrical output in this study differs from that in other studies. Moreover, the multilayer design in this study achieves infarcted myocardium healing and monitoring.

Fig. 5 Schematic illustration of the design and working mechanism of different multilayer TENG (A) published in ACS Nano 2013, 7, 4, 3713–3719. (B) published in ACS Nano 2019, 13, 698-705. (C) in this study; and (D) TENG design and (E) TENG working mechanism published in Nat. Commun. 2016, 7, 12985.

13. Why are the voltage outputs positive and negative (Fig. 2)? Discuss it analytically.

Answer: Thank you for your question. The voltage outputs were measured by connecting the rGO electrode to the positive terminal of a benchtop digital multimeter

(Keithley DMM6500, Tektronix®). Thus, the output voltage is the difference of potentials between the rGO electrodes and the ground ($V_{oc} = P_{rGO} - P_g$) (Fig. 6). The ground has a potential of zero. When the PVDF layer and the PDA-rGO layer contact with each other, opposite triboelectric charges are generated at the contacting surfaces of the two layers with a surface charge density of σ_c . When the PVDF layer and the PDA-rGO layer separate from each other at a gap distance of L , an electric field $E_a = \frac{\sigma_c}{\epsilon_0}$ ($\epsilon_0 =$ dielectric permittivity of air) is established in the air gap between them, resulting a potential difference $V_a = \frac{\sigma_c}{\epsilon_0}L$ in the air gap. To screen out the potential difference, positive charge flow is driven instantaneously from the ground to the rGO electrode, resulting in induced charges in the rGO electrode with a density of σ_I . To drive the positive charge flow, the potential of rGO electrode should be lower than that of the ground. Thus, the voltage output is negative during the separation of the two layers. An electric field E_d is consequently built in the PVDF dielectric layer. The thickness of the PVDF dielectric layer is d . The potential difference in the PVDF dielectric layer $V_d = \frac{\sigma_I}{\epsilon_r}d$ ($\epsilon_r =$ dielectric permittivity of PVDF). To screen out the potential difference, $|V_a| = |V_d| \rightarrow \left| \frac{\sigma_c}{\epsilon_0}L \right| = \left| \frac{\sigma_I}{\epsilon_r}d \right|$. When the PVDF layer and the PDA-rGO layer come into contact again. The gap distance decreases, thus, $|V_a|$ decreases. To maintain $\left| \frac{\sigma_c}{\epsilon_0}L \right| = \left| \frac{\sigma_I}{\epsilon_r}d \right|$, σ_I should decrease. Positive charges flow back to the ground. To drive the back flow of positive charges. The potential of the rGO electrode should be higher than that of the ground. Thus, the voltage output is positive during the contact of the two layers.

Fig. 6 The scheme of the sensing mechanism of the TRI-TENG and the illustration of the electric fields in the TRI-TENG while contracting.

Reviewer #3 (Remarks to the Author):

This manuscript describes the development of a novel strategy for creating an electrically conductive and self-sensing cardiac patch to facilitate the regeneration of injured myocardium after myocardial infarction (MI). The researchers employed a concept based on a trinity triboelectric nanogenerator (TRI-TENG), using reduced graphene oxide (rGO) and biomimetic patterned polydopamine (PDA) to generate suspendable and sandwiched sheets (placed between PDMS spaces) for generating an electric field based on the triboelectric characteristics of these layers. The manuscript presents a wealth of data, both in vitro and in vivo, covering material characterization and the testing of the proposed system's capabilities in small and large animal models to investigate the hypothesis that the proposed triboelectric nanogenerator can stimulate injured myocardium during the contractility cycles and facilitate tissue regeneration. The data demonstrates the effectiveness of the proposed system in myocardial regeneration in both small and large animal models. Additionally, mechanistic studies were conducted using RNA-seq analyses to elucidate the role of specific upregulated pathways and gene signatures that may contribute to enhanced regenerative outcomes. Despite significant amount of data, there are still questions yet to be explored and addressed in this work:

1、 It's important to note that the concept of a triboelectric nanogenerator has been previously utilized by both the authors and other investigators in the field of cardiac regeneration. While the concept itself is not entirely novel, this manuscript introduces a new patch system design built upon this established concept along with extensive amount of data.

Answer: Thank you very much for your comments. Recently, there has been a plethora of research deploying implantable TENGs to cardiac system healthcare (Supplementary Table 1). These implantable TENGs can be categorized into two streams based on their application. They either work as 1) sensors for myocardial abnormality detection or 2) self-powered power sources for pacemakers or therapeutic electrodes. However, our TENG, with its unique double-spacer design, can achieve therapeutic and diagnostic purposes simultaneously. In addition, the TENGs involved in therapeutic purposes

apply the electrical stimulus through an additional electrode. Our double-spacer design incorporates the myocardium as a triboelectric-charge-generating component in TENG, directly building electrical potential on the myocardium without the need for excess electrodes. Moreover, the healing efficacy of electrical stimulus powered by therapeutic TENGs has only been studied *in vitro*, using CMs. Our TENG was applied to both the minipig and rat MI models to assess its efficacy.

Our TENG adopts a unique double spacer design, setting it apart from other TENGs. Multilayer designs have been adopted in TENGs to scale up electrical output while maintaining a constant TENG area (ACS Nano 2013, 7, 3713-3719. ACS Nano 2018, 13, 698-705. Nat. Commun. 2017, 8, 88). However, our TENG distinguishes itself from them by 1) having the first spacer situated on the myocardium enabling the dual function of the PDA-rGO electrode—to generate triboelectric charges as one component in the TENG and to build electrical potential difference on myocardium as a therapeutic electrode; 2) employing the second spacer on top of the PDA-rGO, allowing triboelectrification with another triboelectric layer with greater ability to generate triboelectric charges, which the advantage of electrostatic induction to amplify the electrical potential difference built between the PDA-rGO electrode and the myocardium.

Furthermore, it is the first time mold casting to impart a leaf vein structure to bestow the polyvinylidene fluoride (PVDF) triboelectric layer with a biomimetic leaf vein structure. The leaf vein structure and the PDA coating on the rGO electrode are both nature-inspired surface structures that can enhance the triboelectric effect by increasing the roughness and effective contact areas in a cost-effective manner. Our TENG is the first to adopt nonmetallic rGO and PDA-rGO electrodes as electrode layers in a TENG, whereas other TENGs inevitably utilize metallic electrode layers (Supplementary Table 1).

Supplementary Table 1: The application of Triboelectric Nanogenerator (TENG) in cardiovascular healthcare.

Design	Diagnosis	Therapeutic	Ref.
TENG + rectifier + capacitor + pacemaker	No	Corrects arrhythmia in rat	1
TENG + rectifier + capacitor + pacemaker	No	Corrects arrhythmia in Yorkshire porcine (35 kg)	2
TENG + power manage system + Li-polymer battery + pacemaker	No	Correct arrhythmia in mongrel (25-30 kg)	3
TENG + rectifier + interdigital electrode	No	Increases and unifies beating rates of cardiomyocytes (CM)	4
TENG + interdigital electrode	No	Promotes the maturation of neonatal CM and increases intracellular Ca^{2+} level, Ca^{2+} transient rate, and Ca^{2+} peak amplitude of CM	5
TENG + power manage unit (rectifier + capacitor + switch) + implantable transmitter + receiving coil + oscilloscope	Detect cardiac contractility and heart rate in Yorkshire porcine (30 kg) wirelessly	No	6
Arterial pressure catheter + TENG + lead wires + data acquisition system	Detect heart rate, respiratory rates, blood pressure, and the velocity of blood flow in Yorkshire porcine (30 kg)	No	7
TENG + lead wires + data	Detect endocardial	No	8

acquisition system	pressure and cardiac arrhythmias in Yorkshire porcine (30 kg)		
TENG + catheter encapsulated carbon nanotube + electrometer	Detect heart rate and monitor heart motion in Sprague Dawley rats (250 g)	No	9

1. Zheng, Q. et al. In vivo powering of pacemaker by breathing-driven implanted triboelectric nanogenerator. *Advanced materials* 26, 5851-5856 (2014).
2. Ouyang, H. et al. Symbiotic cardiac pacemaker. *Nature communications* 10, 1821 (2019).
3. Ryu, H. et al. Self-rechargeable cardiac pacemaker system with triboelectric nanogenerators. *Nature communications* 12, 4374 (2021).
4. Jiang, W. et al. Fully bioabsorbable natural-materials-based triboelectric nanogenerators. *Advanced Materials* 30, 1801895 (2018).
5. Zhao, L. et al. Promoting maturation and contractile function of neonatal rat cardiomyocytes by self-powered implantable triboelectric nanogenerator. *Nano Energy* 103, 107798 (2022).
6. Zheng, Q. et al. In vivo self-powered wireless cardiac monitoring via implantable triboelectric nanogenerator. *ACS nano* 10, 6510-6518 (2016).
7. Ma, Y. et al. Self-powered, one-stop, and multifunctional implantable triboelectric active sensor for real-time biomedical monitoring. *Nano letters* 16, 6042-6051 (2016).
8. Liu, Z. et al. Transcatheter self-powered ultrasensitive endocardial pressure sensor. *Advanced Functional Materials* 29, 1807560 (2019).
9. Zhao, D. et al. Eco-friendly in-situ gap generation of no-spacer triboelectric nanogenerator for monitoring cardiovascular activities. *Nano Energy* **90**,

106580 (2021).

2、 The authors have not investigated the degradation of the PDMS and its potential influence on the functionality of the proposed system in vivo. Most of the testing has been conducted over a four-week period, and it remains unclear whether the proposed system, once the PDMS degrades, will remain functional for a longer duration, such as 12 weeks. This is an important consideration, as many previous translational studies on myocardial infarction (MI) have assessed the efficiency of engineered patches for more extended periods, typically up to three months.

Answer: Thank you for your valuable comments regarding the long-term functionality and degradation of the encapsulation material in our proposed system. We apologize for any confusion caused by the incorrect labeling of the material as PDMS; it is indeed Ecoflex. We acknowledge the importance of understanding how the degradation of Ecoflex impacts the performance of the triboelectric nanogenerator (TENG) over extended periods. As reported in the literature, the weight loss of Ecoflex is less than 5% within 60 days (ISME Commun 2023, 3, 67) and less than 10% within 200 days (Polymers (Basel) 2022, 14, 1515). Ecoflex would barely degrade over approximately 12 weeks, ensuring the stability of the TENG's function. We will clarify this point and provide a more detailed analysis of the material's degradation and its implications for long-term functionality in the revised manuscript. Thank you for helping us improve the clarity and thoroughness of our study.

3、 The data presented in Figure 2 need to be tested and generated for long periods of cycles. Perhaps an automated stretching/flexing set up would help the authors to test the capabilities, durability and reproducibility of the proposed system for long duration of cycles.

Answer: Thank you for your suggestion. A 1000-cycle test has been performed to test the capabilities, durability, and reproducibility of the proposed system for long duration of cycles (as shown below). The amplitude of output voltage remains relatively stable even after a substantial number of compression and relaxation cycles.

Fig. 1 Voltage output of a TRI-TENG measured at different compression cycles.

4、 In their gene expression analyses, the investigators have shown upregulated pathways using bulk RNA sequencing. However, a question that remains unanswered is why the authors did not employ single-cell RNA sequencing to examine how the phenotypic signatures of cardiomyocytes, cardiac fibroblast cells, and potentially endothelial cells influence the tissue regeneration process as well as vascularization when exposed to the proposed patch.

Answer: Thank you for your insightful suggestion. In this study, we investigated the change of genes level in different regions of infarct heart using whole-transcriptome RNA sequencing. The result revealed that TRI-TENG or PDA-rGO/Ecoflex transplanted hearts showed altered gene expression which more effectively regulated cardiac contraction, energy metabolism, ECM-receptor interaction, and inflammatory response. We acknowledge that while our current study utilized bulk RNA sequencing to explore the gene expression changes in different regions of the infarcted heart, this approach does provide an average expression level across a mixed cell population and may overlook the transcriptome heterogeneity of individual cells (Cells, 2023, 12(18): 2295).

Although we achieved the desired gene changes using whole-transcriptome RNA sequencing, we agree that single-cell RNA sequencing is a powerful technique that could provide a more detailed understanding of how different cardiac cell populations

including CMs, cardiac fibroblasts, vascular smooth muscle cells, and endothelial cells contribute to the tissue regeneration process and vascularization. Due to constraints in funding and resources, we were unable to include this method in the current study. However, we recognize its importance and, as such, plan to incorporate single-cell sequencing in future research to gain a deeper insight into the cellular mechanisms driving myocardial infarction (MI) repair. Your feedback is invaluable, and we thank you for helping us enhance the scope and depth of our research.

5.- The sensing capabilities of the proposed patch were demonstrated in only one of the studies, using the Langendorff setup. Therefore, it raises the question of why the authors did not evaluate the self-sensing capabilities over the long term, particularly using the mini-pig studies.

Answer: Thank you for your advice. In addition to using the Langendorff setup, we also tested the sensing capabilities of the proposed patch in animal trials with rats and minipigs as illustrated in Fig. 3 A, F in the revised manuscripts. In these trials, the animals were anesthetized, performed ventilation and then carried out thoracotomy for sensing experiment. TRI-TENG patch was transplanted onto the heart and connected to an external sensing device via a wire to transmit the voltage signal.

In the sensing experiment's long-term detection, we encountered practical challenges in maintaining the integrity of the wire connection due to uncontrollability of rat activities. The wire usually gets destroyed in 3–7 days. The limitation has indeed restricted our ability to evaluate the self-sensing capabilities over the long term. We are actively working on improving the device design to facilitate long-term monitoring and minimize interference from animal activity. Your suggestion is invaluable, and it will certainly guide our future research efforts to develop more robust and reliable long-term monitoring solutions. Thank you for helping us improve the rigor and applicability of our work.

Reviewer #4 (Remarks to the Author):

This manuscript describes a E-cardiac patch of miniature self-powered biomimetic trinity triboelectric nanogenerator (TRI-TENG) for sensing and repairing infarcted myocardium. The authors demonstrated that this device improved the electroactivity of

the infarcted heart, including the excitation-contraction coupling, Ca²⁺ transient and action potential (AP) propagation. Furthermore, the authors successfully revealed the patch could also wirelessly monitor electrocardiosignal to the mobile device for diagnosis. RNA sequencing analysis from rat hearts revealed that E-cardiac patch mainly regulated cardiac muscle contraction-, energy metabolism-, vascular regulation-related mRNA expression in vivo. While the in vivo characterization and presentation of the work is good it is not clear what the impact of the E-cardiac patch is. It is critical for the implantable device to have degradable properties for short-term service in vivo. Thus, novelty is limited. In addition, more explains are also needed to strengthen authors' statements.

1. The nanogenerator can convert mechanical energy into electrical signal/electricity. Notwithstanding, it is unrealistic for a single device to achieve both functions (sensing and electrical stimulation repair) at the same time. Are you conducting separate experiments to verify the sensing and treatment functions of the E-cardiac patch? Or does a single patch have both sensing and treatment functions?

Answer: Thank you for your insightful comment regarding the dual functionality of the TRI-TENG in our E-cardiac patch. A single patch has both sensing and treatment functions at the same time. However, measuring the treatment and sensing functions experimentally at the same time poses a challenge. As we've discussed in Rev.2 Q2 and Rev.3 Q5, our Bluetooth module is currently in prototype stage and is not sufficiently miniaturized for implantation in the body. To measure the voltage output, the in-body TENG needs to be connected to the out-body Bluetooth module through a cross-body wire. Due to the uncontrollability of rat activities, the cross-body wire would be destroyed within 3-7 days, posing a challenge to evaluate sensing function. In our study, these functions were indeed investigated separately to ensure the accuracy and effectiveness of each.

For the sensing experiments, we utilized a TRI-TENG with a wired connection to monitor and record the electrical signals generated by the heart's mechanical motion. On the other hand, for the therapeutic aspect, we employed a wireless TRI-TENG as a patch to facilitate the repair of MI. This approach allowed us to observe significant

reparative effects in both rat and mini-pig MI models.

We acknowledge the importance of clearly distinguishing between these two modes of operation and appreciate your bringing this to our attention. Your feedback is crucial for us to communicate our methods and results more effectively. We are committed to further exploring and refining these functions to enhance the potential clinical applications of our E-cardiac patch. Thank you once again for your valuable contribution to our research.

2. Due to myocardial cell necrosis at the site of myocardial infarction, cardiac systolic and diastolic function will decrease in this area. Therefore, can the E-cardiac patch still convert cardiac biomechanical energy into electrical energy well in the area of myocardial infarction? The authors need to add a follow-up experiment to demonstrate the actual electrical performance of the device in the myocardial infarction region compared to normal heart tissue.

Answer: Thanks for your comment. According to your request, we have conducted additional *ex vitro* animal experiment to evaluate the electrical properties of the E-cardiac patch on both normal and injured myocardium. We prepared a smaller size TRI-TENG (0.6 cm in diameter) compared to the size described in the manuscripts (0.8 cm in diameter), which can be completely placed on the infarct region and is beneficial to evaluating the effect of reduced cardiac contractile motion in the infarct region on TRI-TENG' function.

In our experiment, rat hearts were quickly excised, and perfused retrogradely using Langendorff apparatus to preserve vitality. Electrocardiogram (ECG) electrodes were positioned on the right atrium and left ventricle respectively through II leads method, and the amplitude of ECG signal indicates the field potentials between the electrodes, as previously reported (Theranostics. 2021, 6, 11(8), 3948-3960). After the TRI-TENG being transplanted into the left ventricle of the heart, it can convert the biomechanical energy of the left ventricular region into electrical energy, which generates electrical stimulation to the heart, and the changes of field potential would be recorded by the ECG

We firstly recorded the ECG signals of the heart after the TRI-TENG being transplanted into the left ventricle under the normal condition, and found that the field potential amplitude of the heart was significantly increased after TRI-TENG transplantation (Fig. 1C). This result showed that the changes of field potential induced by TRI-TENG can be represented by the changes of ECG amplitude. Subsequently, cardiac ischemia was produced by ligation of the left anterior descending (LAD) artery, which caused an elevation in the ST segment of the ECG (Fig. 1A) and then TRI-TENG was transplanted to the ischemic region of the left ventricle (Fig. 1B). As shown in the Fig. 1C, the field potential decreased by $\sim 0.75\text{mV}$ after ligation, while the field potential increased by $\sim 1.94\text{mV}$ after TRI-TENG transplantation. The results showed a significant improvement in field potential amplitudes of the heart after the TRI-TENG transplantation under the ischemic condition, indicating that the reduced cardiac contractile motion can also induce electrical energy generation by TRI-TENG (Fig. 1C). Besides, we observed that the changes of the field potential amplitude (Δ field potential amplitude) induced by the TRI-TENG after being transplanted into the ischemic heart were lower than those after being transplanted into the normal heart (Fig. 1C). These evidences prove that the reduced mechanical motion can affect the values of electrical output of the TRI-TENG, while the TRI-TENG is still able to convert the reduced mechanical energy of the infarcted myocardium into electrical energy.

Fig. 1 (A) Representative ECG traces of the normal heart and the ischemic heart without transplanted TRI-TENG. (B) Profile display of the ECG electrode placement for langendorff-perfused hearts, where TRI-TENG was transplanted into the ischemic region. The red arrow indicates the site of ligation. (C) Statistics analysis of the field potential amplitude obtained from Langendorff-perfused normal and ischemic rat hearts before and after TRI-TENG transplantation. (n=3 each. The data were presented as mean \pm SD. *P < 0.05, **P < 0.01.)

3. The manuscript lacks any information regarding the reproducibility of the findings. How many animals did you use? How many measurements of voltage, current, and charge were made? Why choose two animal models, swine and rat? There is no necessary connection between the electrical performance in large animals and the actual effects in small animals.

Answer: Thank you for the special comment. According to your advice, we have added the number of repetitions for all experiments to ensure the rigor of experiments. These additions have been highlighted in blue in the revised manuscript.

Regarding your query about the choice of animal models, the rat myocardial

infarction (MI) model is frequently used due to its lower cost and well-established surgical methodology. This model is particularly useful for verifying the effects of conductive cardiac patches on electrical behavior and heart repair. However, to address clinical applicability, a more relevant animal model is necessary. It is widely recognized that the anatomical structure and physiological function of porcine hearts closely resemble those of human hearts. Therefore, we employed the porcine model as a large animal model to further investigate the electrical behavior of cardiac patches. This approach ensures that our findings are more directly applicable to potential human clinical applications.

4. One of the key elements of this work is the novel myocardial patch. The authors need to present a physical picture of the E-cardiac patch. In addition, the authors should clarify the actual dimensions of the devices used in the different experiments.

Answer: Thank you for the special comment. According to your suggestion, we have added the physical pictures of the E-cardiac patch in the supplementary Fig. 6 and also shown in the following Fig. 6. In Fig. 6A, the E-cardiac patch, used in cell and rat experiments, measures 8 mm in diameter. Meanwhile, the TRI-TENG array, employed in large animal experiments, measures 40 mm by 40 mm, as shown in Fig. 6B. We have added these detailed descriptions to the revised manuscript for clarity.

Fig. 2 The physical pictures of the E-cardiac patch. (A) Round TRI-TENG patch employed in cell and rat experiments. Scale bar: 0.5 cm. (B) TRI-TENG array used in large animal experiment. Scale bar: 1 cm.

5. The edges of patches were sutured on the epicardium of the border of the infarcted myocardium with 7-0 polypropylene sutures. Does the suturing process have a negative

impact on the myocardial infarction area? In addition, myocardial infarction repair is a relatively short-term treatment process. After the E-cardiac patch repairs MI, will the device be removed through a second surgery?

Answer: Thank you for your valuable feedback. The suturing process of epicardial patches on the border of infarcted myocardium is a delicate procedure aimed at repairing MI. While suturing with 7-0 polypropylene sutures might slightly affect the myocardium tissue, it does not impact the structure and function of the tissue, as indicated in our previous report (Nat. Biomed. Eng. 2021, 5(10), 1157-1173) as well as many others reports (Acta Biomater. 2020, 101, 206-218. J Thorac Cardiovasc Surg, 2014, 148(3), 1090-1098). This suggests that while there is a minor immediate effect due to the suturing, it doesn't result in a negative impact on the MI area over time.

Regarding the long-term presence and removal of the cardiac patch, ideal patches should degrade without causing toxicity. However, most current conductive cardiac patches do not completely degrade after MI repair. In our previous report (Nat. Biomed. Eng. 2021, 5(10), 1157-1173), observations have shown that these patches can remain on the heart for extended periods, such as six months post-transplantation, without migrating to other organs or causing damage to major bodily systems *in vivo*. Therefore, according to the above evidence, long-term retention of the E-cardiac patch *in vivo* does not cause significant toxicity and did not need to remove through a second surgery.

6. Fig. 1A- the third column shows the healing mechanism that an electrical potential built between the PDA-rGO electrode and the myocardium upon the contraction and relaxation of the myocardium. What does each rectangular unit represent? The authors need to rephrase it clearly.

Answer: Thank you very much for pointing out the lack of clear legend. We have made a new figure showing the healing mechanism with explicit explanation for the meaning of each rectangular.

Fig. 3 Working mechanisms of the TRI-TENG as a healing patch.

7. Fig. 2A. The PDA coating on the PDA-rGO electrode further increased the open-circuit voltage, short-circuit current, and transferred charges to 21.98 mV, 2.23 nA, and 0.22 nC, respectively. The authors should further explain the specific reasons.

Answer: Thank you for your comments. As shown in Supplementary Fig. 1C, the PDA attached to rGO forms granules, which significantly increases the roughness of PDA-rGO electrode. The charge density of the generated triboelectric charges on the surface of the PDA-rGO electrode is higher than that on the rGO electrode. The resulted open-circuit voltage, short-circuit current, and transferred charges are consequently higher.

8. Why are the structures involved in device performance characterization in vitro not the same as in vivo? In vivo testing of nanogenerators will be seriously affected by environmental interference. The electrical output of the device in the body is in the order of mV and nA. How exactly did the authors measure it?

Answer: Thank you for your comments. In the in vitro device performance characterization, we tested the performance of three different double spacer TENGs, employing various materials for the dielectric layer and electrode layer that contacts the myocardium. We found that TENG with a patterned PVDF dielectric layer and a PDA-rGO electrode layer exhibited the best performance. Thus, we utilized the PDA-rGO and patterned PVDF to fabricate TENG for in vivo testing. In the rat model, the testing TENGs were the same size as those used for in vitro testing. However, the heart of a

minipig is considerably larger than that of a rat. If TENG with only increased area is applied to the heart of minipig, the TENG cannot have a perfect attachment with the large and curved surface of minipig's myocardium. Furthermore, the PDA-rGO electrode of the TENG with only increased area cannot establish sufficient contact and separation with the minipig's myocardium. To ensure the fitness to large and curved surface areas, as well as to establish sufficient contact and separation, 9 square-shaped TRI-TENG has been assembled in a 3×3 array design with each neighboring connected rGO electrodes connected and neighboring connected PDA-rGO electrodes.

For the in vitro test. The benchtop digital multimeter (Keithley DMM 6500, Tektronix®) was used to measure electrical outputs. The Keithley DMM 6500 has a sensitivity of 100 nV in voltage measurements and 10 pA in current measurements. The pocket multimeter (Pokit Meter, Pokit Innovations®), which is an all-in-one multimeter, oscilloscope, and logger, was utilized to demonstrate the wireless sensing capability of the TRI-TENG by measuring voltage outputs in the body. The Pokit Meter can measure voltage down to 10 mV. The output current was not measured in the body.

9. By watching the video in the supplementary material, we found that the device is still relatively large and hard. Are the same devices used in long-term in vivo experiments?
Answer: Thank you for your insightful comments regarding the dimensions and rigidity of the TRI-TENG device used in our experiments. We appreciate your concern about the device's suitability for long-term in vivo use. The video is mainly to demonstrate the wireless sensing function of the TRI-TENG. In the process of preparing the TRI-TENG with wire, we used conductive silver glue, which increased its hardness. The wire is added to measure the sensing function of the TRI-TENG. We used the TRI-TENG without wire in the long-term animal experiments. The Young's modulus of the TENG without wire is 640.70 ± 71.07 Kpa (Fig. 4), which aligns with most the scaffolds used in MI management (Bioact. Mater. 2021, 6, 2198-2220). Your comments are very meaningful, and we will improve them in the future research.

In our study, we carefully considered the size and flexibility of the TRI-TENG device to match the range of infarcted areas in different animal models. Specifically,

for the rat *in vivo* experiments, the dimensions of the TRI-TENG were tailored to fit the infarcted area typically observed in rats. Moreover, for the minipig experiments, we assembled an array of unit TRI-TENGs in series to adequately cover the infarct size of a porcine heart. We also emphasize that our cardiac repair studies, which involved both rats and minipigs, were conducted over a period exceeding one month. These studies yielded positive experimental outcomes, indicating the device's efficacy and tolerance in a long-term *in vivo* setting. Based on these considerations and results, we are confident that the dimensions and physical properties of the TRI-TENG devices are within a reasonable range for effective use in long-term *in vivo* experiments. Our findings suggest that the same devices can be successfully employed for extended periods without compromising the experimental objectives or the well-being of the animal models.

We hope this explanation addresses your concerns and provides clarity on the suitability of our devices for long-term *in vivo* experiments.

Fig. 4 Mechanical properties of the TRI-TENG (A) Strain-stress curve and (B) Young's modulus. The data were presented as mean \pm SD. n=3.

10. Are you running wires from the electrodes to test the electrical output of the E-cardiac patch? Does the device still have a wire structure during the process of repairing myocardial infarction? The authors need to further clarify how the E-cardiac patch works *in vivo*.

Answer: Thank you for your thoughtful inquiry regarding the wiring and operational mechanism of the E-cardiac patch *in vivo*, particularly during MI repair. To clarify, the E-cardiac patch is designed with the capabilities of conduction, sensing, and transforming biomechanical energy into electrical stimulation (healing). The E-cardiac

patch achieves the three functions simultaneously. However, we still need wires to evaluate the sensing function of our TENG due to the immaturity of our Bluetooth module. As we have discussed in Rev.2 Q2, Rev.3 Q5, and Rev. Q1, a cross-body wire, susceptible to destruction within 3-7 days due to uncontrollable rats' activities, is connected to the rGO layer for sensing function. The healing and sensing functions were tested separately to ensure their accuracy and effectiveness in our study.

Thus, there was no wire connected to the rGO layer of the E-cardiac patch during the in vivo application for MI repair in animal models. Our E-cardiac patch adopts a unique double-spacer design, incorporating the myocardium as one component of the TENG. The first spacer makes a gap between the PDA-rGO electrode layer and the myocardium. The PDA-rGO electrode layer contacts with the myocardium upon the relaxation of the heart, resulting in positive charges on the myocardium and negative charges on the PDA-rGO electrode layer. The separation of the two layers upon the contraction of the heart builds an electrical field between the PDA-rGO electrode layer and the myocardium. The E-cardiac patch can directly build an electrical field on the myocardium without excess electrodes. Consequently, our approach facilitates effective repair of MI, as evidenced by the improved cardiac function observed in our animal models.

We hope this explanation provides a clearer understanding of the E-cardiac patch's function during MI repair and addresses your concerns regarding the use of wires in its application.

11. Minor comments

i. >Figure 2B: Please modify the expression of the ordinate - "Current (nA) (nA)"

ii.>The picture clarity of the manuscript needs to be improved.

Answer: Thank you for your comments. We have modified the ordinate "Current (nA) (nA)" to "Current (nA)" of Fig. 2 in the revised manuscript. Also, the pictures in the manuscript are compressed in order to reduce the storage of the document. We have uploaded the artwork to the submission system.

REVIEWER COMMENTS

Reviewer #2 (Remarks to the Author):

The authors have properly addressed my comments.

Reviewer #3 (Remarks to the Author):

The authors have addressed the majority of the concerns. The degradation of the EcoFlex has not been still discussed in the revised manuscript. In addition, the Figures provided in the response page could be added as supplementary figures for further clarification of the manuscript.

Reviewer #4 (Remarks to the Author):

I have carefully reviewed the detailed responses provided by the authors. The presented E-cardiac patch is an improvement. However, I still think that a single device combined sensing and stimulation functions is not very practical in the diagnosis and treatment of myocardial infarction, although the idea is good. First, the sensing unit and the stimulation unit will interfere with each other and affect their respective functions, especially the flexible TENG device. Secondly, the multi-functional device will make the system more complex and reduce the stability of the device. In addition, the purpose of the sensor is to feedback effective information in a timely manner for adjusting the stimulation mode, but the sensing and electrical stimulation data currently presented by the authors are independent of each other. If the authors can combine sensing and stimulation organically and provide sufficient data, I believe it will be a good research work. In addition, more explains are needed to strengthen authors' statements from the responses. The authors have added ex vitro animal experiment to evaluate the field potential amplitude obtained from Langendorff-perfused normal and ischemic rat hearts before and after TRI-TENG transplantation. Why didn't the authors directly measure the voltage and current of the device? What electrode materials are used to connect the various units of the TRI-TENG array device?

REVIEWER COMMENTS

Reviewer #2 (Remarks to the Author):

The authors have properly addressed my comments.

Reviewer #3 (Remarks to the Author):

The authors have addressed the majority of the concerns. The degradation of the EcoFlex has not been still discussed in the revised manuscript. In addition, the Figures provided in the response page could be added as supplementary figures for further clarification of the manuscript.

Answer: Thank you for suggestions. We have added the discussion on the degradation of EcoFlex in the part of “Assembly and characterization of TRI-TENG CCP” of the Results and Discussion section in this revised manuscript. The Figures provided in the previous response have been incorporated into the manuscript now. The description of mechanical properties of the TRI-TENG (Reviewer 4. Q9) has been added as Supplementary Fig. 6. The description of recorded electrical signals and electrocardiogram signals of rat hearts under open-chest and closed-chest conditions (Reviewer2. Q7) has been added as Supplementary Fig. 15, and the description of histological examination and cardiac function evaluation of rat hearts in the normal group and TRI-TENG transplanted group at postoperative week 4 (Reviewer1. Q9) have been added as Supplementary Fig. 17. The corresponding descriptions of these added Figures have also been supplemented in the main text. All modifications have been highlighted in blue in the manuscript.

Reviewer #4 (Remarks to the Author):

I have carefully reviewed the detailed responses provided by the authors. The presented E-cardiac patch is an improvement. However, I still think that a single device combined sensing and stimulation functions is not very practical in the diagnosis and treatment of myocardial infarction, although the idea is good. First, the sensing unit and the stimulation unit will interfere with each other and affect their respective functions, especially the flexible TENG device. Secondly, the multi-functional device will make

the system more complex and reduce the stability of the device. In addition, the purpose of the sensor is to feedback effective information in a timely manner for adjusting the stimulation mode, but the sensing and electrical stimulation data currently presented by the authors are independent of each other. If the authors can combine sensing and stimulation organically and provide sufficient data, I believe it will be a good research work. In addition, more explains are needed to strengthen authors' statements from the responses. The authors have added ex vitro animal experiment to evaluate the field potential amplitude obtained from Langendorff-perfused normal and ischemic rat hearts before and after TRI-TENG transplantation. Why didn't the authors directly measure the voltage and current of the device? What electrode materials are used to connect the various units of the TRI-TENG array device?

Answer: Thank you for your comments. The output of the open-circuit voltage from the rGO electrode of our sensing unit has limited interference with the electrical field built between the PDA-rGO electrode and the myocardium in our stimulation unit due to our unique double-spacer design. Though the strength of the electrical field in the healing unit is dictated by the strength of the electrical field built in the sensing unit, which is ultimately determined by the strength of the heartbeat, the measurement of the voltage output of our sensing unit itself minimally impacts with the healing effect of the stimulation unit. The complexity of the *in vivo* voltage measurement environment does impede the flexibility, healing effect, and stability of our device, however, it can be overcome by integrating our sensing unit with highly integrated flexible electronics for the measurement, which is our focus for future studies.

We have designed the following model study to prove that the sensing unit and stimulation do not interfere with each other. The setup is illustrated in Fig. 1 and Supplementary Movie 1, a sodium alginate (SA) Hydrogel hydrogel that mimics the heart was placed on top of the first spacer layer of our device. The SA hydrogel was subject to cyclic compression to mimic the contact and separation of the PDA-rGO layer with the heart due to heartbeat, the rGO layer of our device was connected to a benchtop digital multimeter (Keithley DMM6514, Tektronix®) for the measurement of the voltage output of sensing unit. Additionally, two electrodes from an

electromyograph (ECG) machine were attached to the hydrogel to measure the electric potential built on the hydrogel by the stimulation unit. As shown in Fig. 1A and Supplementary Movie 1, the field potential on the hydrogel and the voltage output from the rGO electrode can be simultaneously measured by the ECG machine and the multimeter, when the hydrogel is subject to cyclic compression. Moreover, the peak value of both the field potential and the voltage output can remain stable (Fig. 1B, C). Thus, the sensing unit and the stimulation unit don't interfere with each other, without consideration of the complexity of measuring voltage output *in vivo*.

Fig. 1 (A) Schematic illustration of the setup of the mode study. **(B)** Stimulation unit generated electrical field detected by the ECG electrode without the cyclic compression (left) and with the cyclic compression (right). **(C)** The voltage output of the sensing unit measured by the digital benchtop multimeter without the cyclic compression (left) and with the cyclic compression (right).

Indeed, we also directly measure the output voltage of the TRI-TENG to evaluate the electrical properties of the E-cardiac patch on both normal and ischemic rat hearts in the *ex vitro* animal experiment. As shown in Fig. 2, the open-circuit voltage difference (ΔV_{oc}) from the Langendorff-perfused rat heart were 1.294 ± 0.396 mV under the normal condition and 0.164 ± 0.037 mV under the ischemic condition, respectively.

Besides, we observed that the ΔV_{oc} induced by the TRI-TENG after being transplanted into the ischemic heart were lower than those after being transplanted into the normal heart (Fig. 2C). These evidences prove that TRI-TENG can sense weak mechanical motion.

Fig. 2 (A) Representative ECG traces of the normal heart and the ischemic heart without transplanted TRI-TENG. (B) Profile display of the ECG electrode placement for langendorff-perfused hearts, where TRI-TENG with wire was transplanted into the left ventricle. (C) Electrical signal outputs of TRI-TENG recorded from the TRI-TENG-transplanted Langendorff-perfused rat normal heart (left) and ischemic injured heart (right). (D) Statistics analysis of the field potential amplitude obtained from Langendorff-perfused normal and ischemic rat hearts before and after TRI-TENG (without wire) transplantation. (E) Statistics analysis of the ΔV_{oc} based on (C). $n=3$ each. The data were presented as mean \pm SD. * $P < 0.05$, ** $P < 0.01$.

We used a PEDOT: PSS/GelMA hydrogel as solder to connect the various units of

the TRI-TENG array device. The synthesis procedure is outlined below. Firstly, the GelMA was synthesized according to a previously reported technique. In brief, 10 g of gelatin was dissolved in 100 mL PBS buffer at 50 °C under vigorous stirring to get a 10 % (w/v) gelatin aqueous solution, and then 2 mL of methacrylic anhydride (MAA) was added into the gelatin solution. The reaction proceeded at 50 °C for 2 h and the pH of the mixture was kept at 7.4 during the reaction using 10 M NaOH solution. The reaction product was dialyzed against DI water at 50 °C for 3 days and then lyophilized. Next, 100 mg of PEDOT: PSS, 100 mg of GelMA, and 2 mg of Irgacure® were dissolved in 2 mL DI water under vigorous stirring at 50 °C to obtain a homogeneous solution. A small amount (5 - 10 mg) of the precursor solution was placed on each connection point as solder. The precursor solution was then cured under UV for 30 min and air-dried. The related description has been added in the part of “Assembly of TRI-TENG and TRI-TENG array” in the revised manuscript.